# Stem cell niche exit in *C. elegans* via orientation and segregation of daughter cells by a cryptic cell outside the niche

Kacy L Gordon[1]*, Jay W Zussman[2], Xin Li[1], Camille Miller[1], David R Sherwood[2,3]

[1]Department of Biology, The University of North Carolina at Chapel Hill, Chapel Hill, United States; [2]Department of Biology, Duke University, Durham, United States; [3]Regeneration Next, Duke University, Durham, United States

**Abstract** Stem cells reside in and rely upon their niche to maintain stemness but must balance self-renewal with the production of daughters that leave the niche to differentiate. We discovered a mechanism of stem cell niche exit in the canonical *C. elegans* distal tip cell (DTC) germ stem cell niche mediated by previously unobserved, thin, membranous protrusions of the adjacent somatic gonad cell pair (Sh1). A disproportionate number of germ cell divisions were observed at the DTC-Sh1 interface. Stem-like and differentiating cell fates segregated across this boundary. Spindles polarized, pairs of daughter cells oriented between the DTC and Sh1, and Sh1 grew over the Sh1-facing daughter. Impeding Sh1 growth by RNAi to cofilin and Arp2/3 perturbed the DTC-Sh1 interface, reduced germ cell proliferation, and shifted a differentiation marker. Because Sh1 membrane protrusions eluded detection for decades, it is possible that similar structures actively regulate niche exit in other systems.

## Introduction

Stem cells and their associated supportive niche cells are centrally important to development (*Murry and Keller, 2008*; *Zhu and Huangfu, 2013*), homeostatic tissue maintenance (*Biteau et al., 2011*; *Blanpain and Fuchs, 2009*; *Snippert et al., 2010*; *Wilson et al., 2008*), aging (*Rao and Mattson, 2001*), regeneration (*Sánchez Alvarado and Yamanaka, 2014*), cancer (*Reya et al., 2001*), and tissue engineering (*Ruder et al., 2011*). For this reason, the ways in which a niche anchors and signals to its stem cells is an area of active inquiry. Balance must be struck between retaining a sufficient pool of stem cells in the niche and generating a robust population of stem cell descendants that go on to differentiate. Therefore, the question of how stem cell progeny leaves the niche in a regulated fashion is likewise important, but has been less intensely studied. Most of what is known about stem cell niche exit is understood in the context of how the niche retains the self-renewing stem cell daughter rather than how the differentiating daughter escapes the influence of the niche. Examples include stochastic displacement in the mammalian intestine (*van der Flier and Clevers, 2009*), migration in the larval tracheal system of *Drosophila* (*Chen and Krasnow, 2014*), oriented division to a basal lamina in the mammalian epidermis (*Poulson and Lechler, 2010*), and oriented division to the niche cells in the *Drosophila* ovary (*Casanueva and Ferguson, 2004*).

The *C. elegans* germ line is supported by a canonical stem cell niche (*Hubbard, 2007*; *Lander et al., 2012*) called the distal tip cell (DTC). Because of the ease of visualization and experimental manipulation, many general principles have been obtained by analysis of this simple system, including the first demonstration of stem cell niche properties (*Kimble and White, 1981*). The genetics controlling stemness and differentiation are very well understood (*Hubbard, 2007*; *Kimble and Seidel, 2013*), and a cell biological understanding of the system is growing (*Amini et al., 2014*; *Byrd et al., 2014*; *Linden et al., 2017*). The adult DTC has a jellyfish-like

*For correspondence:
kacy.gordon@unc.edu

**Competing interests:** The authors declare that no competing interests exist.

**eLife digest** Stem cells have the rare ability to divide and specialize into the many different types of cells necessary for an organism to survive. For instance, germ stem cells can multiply to produce precursor cells that go on to become eggs or sperm needed for reproduction.

When a stem cell divides, the daughter cells can either remain 'naïve', or start to specialize into a given cell type. In many cases, this decision is strongly influenced by the properties of the environment that surrounds the stem cell. However, in the microscopic worm *Caenorhabditis elegans*, how the daughters of germ stem cells specialize was thought to be a random process, with nearby cells equally likely to specialize or remain naïve.

In this animal, germ stem cells reside in tube-shaped structures called gonads, which are enclosed by a large 'distal tip' cell. In addition, cells known as Sh1 surround the gonad. Here, Gordon et al. tracked dividing germ stem cells in the gonads of live worms. This revealed that both the distal tip cell and Sh1 cells have finger-like extensions that form contacts with the germ stem cells. The fate of dividing germ stem cells is shaped by these interactions. If they touch only the distal tip cell, they remain in a naïve state. However, if they contact both the distal tip cell and an Sh1 cell, one daughter of the stem cell becomes an egg precursor – with the daughter closest to the distal tip cell staying naïve. In fact, germ stem cells that are prevented from contacting Sh1 cells divide less often.

Many other types of stem cells, for example in human skin, are believed to make the decision to remain naïve or undergo specialization randomly. The results from Gordon et al. could provide a roadmap to discover hidden layers of control in other organisms, some of which may be potentially relevant in health and disease.

appearance, with a flattened cell body at the distal end and long trailing processes that extend proximally and enwrap germ cells, including the presumptive stem cells (*Byrd et al., 2014*; *Crittenden et al., 2006*; *Gordon et al., 2019*). The germ line is partially syncytial, with membrane-bound germ cell bodies connected to a common cytoplasmic core (the rachis) by narrow bridges of cytoplasm (*Hirsh et al., 1976*; *Seidel et al., 2018*). Despite the cytoplasmic connections that facilitate the sharing of intracellular fate determinants (*Lee et al., 2016*), the germ line segregates cell fates across its distal-proximal axis, with germ cells undergoing meiosis proximal to the undifferentiated germ cells dividing stochastically in the distal 'progenitor zone'. The progenitor zone is approximately 20 germ cell diameters long (~100 µm) and contains 243 + / - 25 cells in one-day adult animals (*Crittenden et al., 2006*). A subset of the progenitor zone germ cells makes up the germ stem cell pool. The DTC niche expresses the Notch ligands LAG-2 and APX-1 that activate Notch signaling in the germ stem cells (*Henderson et al., 1994*; *Nadarajan et al., 2009*). It has been hypothesized that divisions within the stem cell population simply push daughters out of the niche to eventually differentiate (*Rosu and Cohen-Fix, 2017*), however stem cell progeny breaking contact with the niche have not been visualized. Previous work by our group (*Linden et al., 2017*) suggests that a simple distal-to-proximal model of stem cell position does not take into account the effect of DTC geometry on Notch activation. When the DTC is asymmetrically shaped, only the germ cells closest to it express a Notch reporter, and other equally distal germ cells lack reporter expression, suggesting that close proximity to the DTC rather than distal position defines stem cells (*Linden et al., 2017*). Thus, while downstream effects of localized Notch signaling on germ cell stemness vs. differentiation are well understood, how the niche-stem cell association is organized and how it terminates and releases germ cells to differentiate given the complex and varied niche geometry is not known.

Proximal to the DTC, the remainder of the somatic gonad comprises five pairs of gonadal sheath cells that lie between the germ cells and the gonadal basement membrane (*Hall et al., 1999*). Sheath cell pair 1 (Sh1) covers germ cells in the distal gonad, with one cell on its superficial surface closer to the cuticle and the other on the deep surface closer to the gut (*Hall et al., 1999*). Between the distal boundary of Sh1 and the DTC, a ~ 20 germ cell diameter-long (~100 µm) 'germ line bare region' has been described, in which germ cells are thought to contact the gonadal basement membrane directly, not somatic cells. This germ line bare region makes up the entire progenitor zone

outside of the distalmost cells under the DTC. Evidence for the bare region comes from scanning EM of dissected gonads and cytoplasmic GFP expressed in the gonadal sheath (*Hall et al., 1999*). How the differentiating germ cells navigate the putative somatic cell void between the DTC and Sh1 cells is unclear.

We set out to investigate the process by which germ cells break their contacts with the DTC niche and interact with the other cells of the somatic gonad. We used CRISPR/Cas9 genome editing, live cell time-lapse imaging, and RNAi-mediated gene knockdown to probe the relationships among germ cells, the DTC, and Sh1 sheath cells. Unexpectedly, we discovered that thin membranous processes from the Sh1 cells extend distally and intercalate among undifferentiated germ cells and the processes of the DTC, with soma contacting nearly every germ cell. There is no germ cell bare region. As a result, germ cell divisions take place in three anatomical compartments—in contact with the DTC niche alone, at the DTC-Sh1 boundary, and in contact with Sh1 alone. Divisions on the elaborate interface between the DTC and Sh1 are more frequent and are polarized, with daughters segregating between them, suggesting a mechanism for germ cell niche exit via the first documented asymmetric cell divisions in the *C. elegans* germ line. Sh1 actively grows over and between daughter cells that divide at the interface in an Arp2/3- and cofilin-dependent manner, which strengthens the association between one daughter cell and the sheath. When Sh1 was impeded from growing towards the DTC by RNAi against actin regulators, the dividing populations at the boundary and under Sh1 were reduced and a marker of differentiation was shifted in the same direction as the Sh1 cell, suggesting that distal Sh1 cell processes assist stem cell niche exit. We conclude that Sh1, a cell neighboring the niche, is an overlooked regulator of adult stem cell niche dynamics.

## Results

### The stem cell niche is closely apposed to Sh1

To investigate how the elaborate DTC germ line stem cell niche interacts with other cells in the gonad we used CRISPR/Cas9-mediated genome editing (*Dickinson et al., 2015*; *Dickinson et al., 2013*) to endogenously tag cell membrane-localized gap junction proteins (INX-8 and INX-9) that were previously found to be expressed in the DTC and the gonadal sheath cells (*Starich et al., 2014*). The *inx-8* and *inx-9* genes are genetically redundant with both single mutants fertile and the double mutant sterile (*Starich et al., 2014*). As innexins have not been endogenously tagged with genetically encoded fluorophores previously, we used this *inx-9(ok1502)* mutant allele to ascertain function of the *inx-8* genome-edited alleles. We found that a homozygous C-terminal-tagged *inx-8:: mCherry* in an *inx-9(ok1502)* mutant background was sterile. However, a homozygous N-terminal-tagged *mKate2::inx-8(qy102)* in this mutant background was fertile (*Figure 1—figure supplement 1A*). We used the sequence similarity between the genes to tag the *inx-9* locus in the same N-terminal position. The atomic structure of another *C. elegans* innexin, INX-6, has been determined (*Oshima et al., 2016*); the N-terminus of *C. elegans* INX-6 is extracellular, flexible, and located in the pore of the gap junction. If INX-8 and INX-9 have the same structure, this means the fluorescent tags (from multiple monomers in a hexameric hemichannel) are linked to extracellular extensions from the pore of the gap junctions formed by tagged INX-8 or tagged INX-9.

Upon coexpression of each of these tagged innexins with a DTC membrane marker and examination by live-animal confocal microscopy, we observed expression in the DTC and in the gonadal sheath. Unexpectedly, we found that the long cellular processes of the DTC niche cell were directly bordered by membranous extensions from Sh1 (*Figure 1A and B*), contrary to the accepted model of gonad anatomy with a bare region separating DTC and Sh1 (*Figure 1A*, *Lints and Hall, 2018*). We verified this pattern with another endogenously tagged protein that localizes to the membrane, integrin α subunit INA-1::mNeonGreen (*Jayadev et al., 2019*; *Figure 1—figure supplement 1B*) and by crossing our mKate::INX-8 strain to a strain expressing cytoplasmic GFP (*tnIs6(lim-7p::GFP)* [*Hall et al., 1999*]) to reveal that the Sh1 membrane protrusions can extend farther distal than the brightest cytoplasmic GFP signal in the cell (*Figure 1—figure supplement 1C*). A revised model positions the DTC and Sh1 intercalating among the distal germ cells in the stem cell zone (*Figure 1C*). Unlike other cells in *C. elegans*, including even elaborately branching neurons (*Chisholm et al., 2016*; *Smith et al., 2010*; *Zou et al., 2015*), the extensive DTC-Sh1 interface is variable among individuals (*Figure 1—figure supplement 1D* and *Figure 1—figure supplement 1E*)

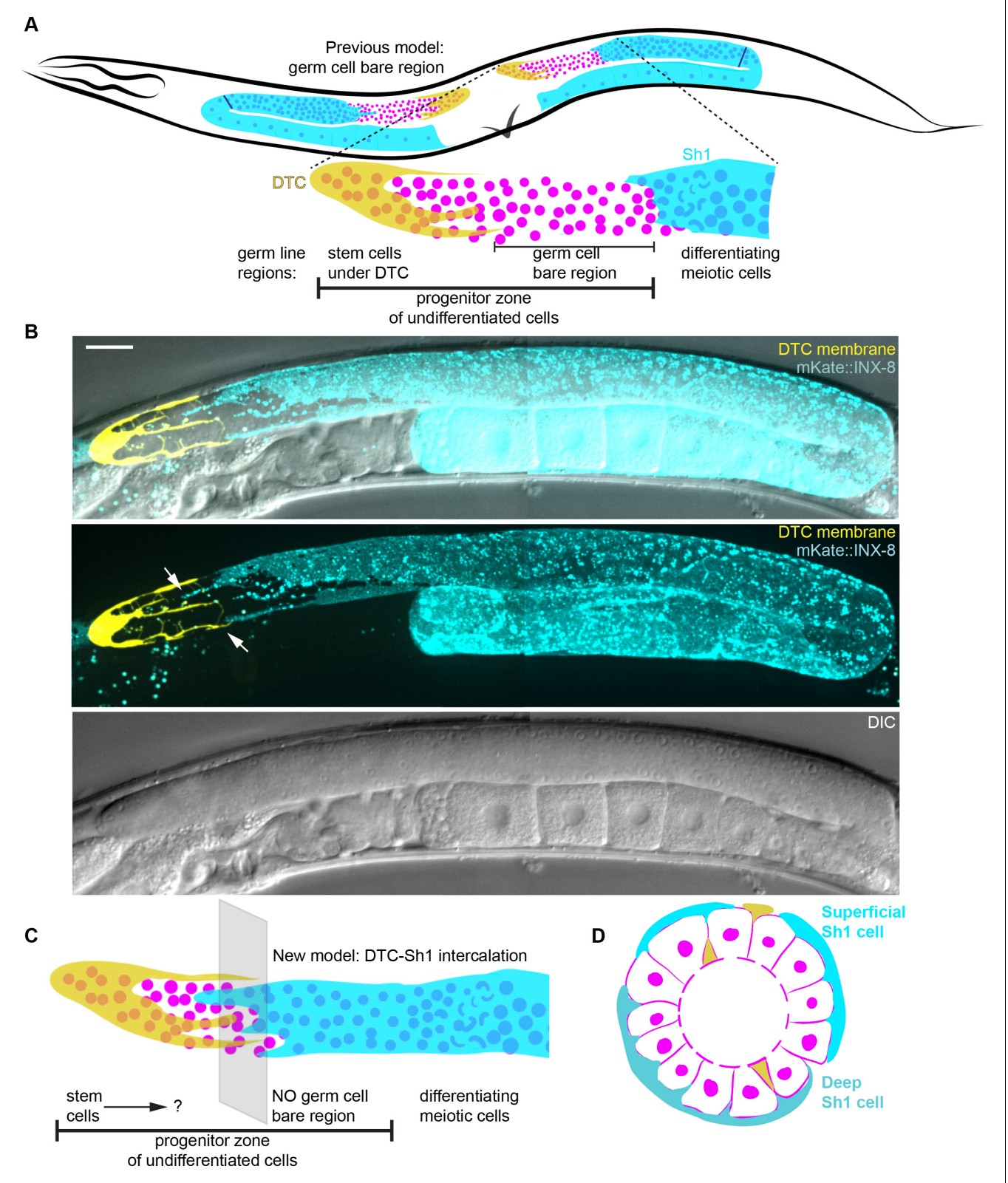

**Figure 1.** Sh1 is present in the distal gonad and intercalates with the processes of the DTC. (**A**) Schematic depiction of the previous model of gonad morphology of *C. elegans* hermaphrodite adults with a germ cell 'bare region' (after [***Lints and Hall, 2018***]). Germ cell nuclei shown in pink, DTC in yellow, sheath in cyan (with Sh1 proximal boundary represented by line near bend). Regions of germ line shown in blow-up below, after (***Crittenden et al., 2006***). (**B**) Micrographs showing the gonad in DIC with fluorescent overlay (top) of the DTC (yellow, *lag-2p::mNeonGreen::PLC$^{\delta PH}$*)

*Figure 1 continued on next page*

*Figure 1 continued*

and Sh1 (cyan, *mKate2::inx-8*), fluorescence only (middle, projection from rachis up through surface of gonad), and DIC only (bottom, single slice through rachis). Images taken at 400x magnification, composite of two tiled fields of view. Scale bar 20 μm. (C) New model based on our observation that Sh1 and the DTC intercalate in the distal gonad. With the DTC closely bordered by an adjacent somatic Sh1 cell, the delineation of the stem cell population is unclear. DTC in yellow, Sh1 in cyan, and germ cell nuclei in pink. (D) Model of cross section of gonad, showing deep Sh1 cell in dark turquoise in addition to superficial Sh1 cell in cyan, DTC processes both superficial and deep in yellow, and germ cell membranes and nuclei in pink. Central void is the rachis. See also *Figure 1—figure supplement 1*.

The online version of this article includes the following figure supplement(s) for figure 1:

**Figure supplement 1.** Endogenously tagged proteins label the somatic gonad, which has significant inter-individual variation and no 'bare region'.

and uneven around the circumference of the gonad (*Figure 1D* shows superficial and deep Sh1 cells above and below the germ line, respectively). Given its juxtaposition with the DTC, we reasoned that the cryptic neighboring Sh1 cells could be in a position to interact with the stem cell niche and potentially with the stem cells themselves.

## Sh1 migrates with the DTC during gonad development

We next examined the relationship of the DTC with the Sh1 cell during development of the mature niche to determine how this elaborate interface is formed. *C. elegans* gonad arm formation entails long-distance postembryonic migration of the two DTCs as the germ cells proliferate behind them, with the entire gonad encased in a growing basement membrane (*Sherwood and Plastino, 2018*). The anterior and posterior DTCs are born from the second division of the gonad primordium cells Z1 and Z4. These migratory leader cells initiate gonad outgrowth in the L2 larval stage (21 hr post-hatching at 20°C). The DTCs migrate in opposite directions along the ventral body wall while the Z2 and Z3 germ line progenitors proliferate. During the L3 larval stage (27 hr post-hatching), the two pairs of Sh1 cells are born (*Kimble and Hirsh, 1979*; *McCarter et al., 1997*). Shortly after, the DTCs turn (30–33 hr) first to the dorsal body wall and then back to migrate to the midline where they come to rest in L4 animals (45 hr, see *Figure 2A*, timing after [*Lints and Hall, 2018*]). The cell body of Sh1 has not been observed during development, owing to its extreme thinness and uneven borders, but it is thought either to migrate over or be dragged by the growing germ line (*Kimble and Hirsh, 1979*). In the adult, its edge is thought to fall approximately midway down the distal branch of the gonad arm (*Hall et al., 1999*; *Killian and Hubbard, 2005*).

To visualize the Sh1 cell during gonad formation, we examined mKate2::INX-8, which was expressed in the DTC and Sh1 cell membranes during gonadogenesis, along with a DTC membrane marker (*Figure 2B*). We found that the larval Sh1 cells were thin, elongated, irregular cells that extended away from the midline over the germ cells and reached towards the DTC with at least one long, thin projection (*Figure 2B*, arrows); the DTC sent at least one reciprocal projection back to touch Sh1 (n = 27/28 gonad arms have DTC and Sh1 in apparent contact). These thin projections persisted as the forming gonad made its first (n = 16/17) and second (n = 10/10) turns (*Figure 2B*). By the end of L4, the Sh1 came to rest with a smooth distal boundary approximately one germ cell diameter from the DTC, which typically extended one or more short, thin projections that touched or nearly touched the distal Sh1 boundary (n = 14/17 within one germ cell diameter at the end of migration). We conclude that the DTC and Sh1 share an intimate association during gonad migration in the larva (schematic *Figure 2C*).

## During the L4/Adult Transition, the DTC and Sh1 Elaborate Dramatically

We continued our time-course analysis to determine when the DTC and Sh1 intercalate. The sharp distal boundary between the DTC and Sh1 was dramatically remodeled beginning at the L4/adult transition, coincident with the elaboration of the DTC (n = 4/4, *Figure 2—figure supplement 1*). The DTC transformed from its high, narrow larval cap morphology to a broad, thin, distended structure with proximal projections. As the DTC elaborated, Sh1 filled in the gaps between extending DTC processes (n = 13/14 young adults, *Figure 2B* and *Figure 2D*) instead of residing in the middle of the distal arm of the gonad as was previously thought (*Hall et al., 1999*; *Killian and Hubbard, 2005*; *Korta and Hubbard, 2010*). It has been hypothesized that the distal boundary of Sh1 would

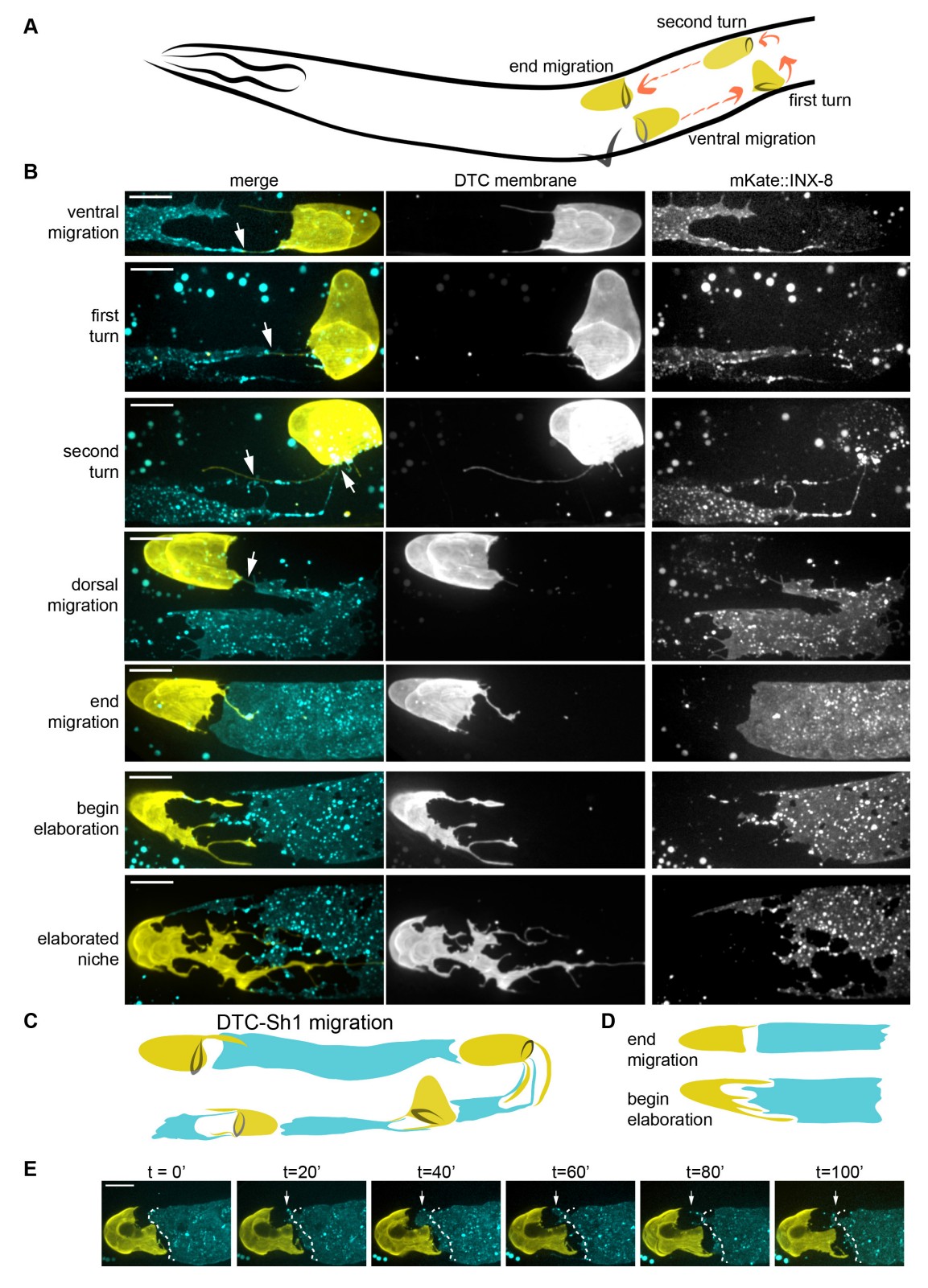

**Figure 2.** The DTC and Sh1 cells are closely apposed throughout development. (**A**) Schematic depiction of *C. elegans* DTC (yellow) migration path (orange arrows) during development, timing after (*Lints and Hall, 2018*): ventral migration begins at 21 hr post hatching; first turn at 30 hr post hatching, second turn at 33 hr post hatching, and end migration at 45 hr post hatching. (**B**) Time course analysis of somatic gonad elongation. DTC (yellow, *lag-2p::mNeonGreen::PLC$^{\delta PH}$*) and Sh1 (cyan, *mKate2::inx-8*, also detected in larval DTC). The two cells are in contact or within one germ cell

*Figure 2 continued on next page*

*Figure 2 continued*
diameter throughout development (white arrows). (C) New model of gonad development including Sh1 cell (only one of pair shown). (D) Schematic showing of the DTC-Sh1 boundary transformation between the end of migration and beginning of elaboration. (E) Stills from a time-lapse movie (*Figure 2—video 1*) of same strain shown above. Dashed white line shows position of distal Sh1 at time 0 in all subsequent panels; note remodeling of Sh1 back and forth to extend and close this distance (arrows). All scale bars 10 µm. See also *Figure 2—video 1* and *Figure 2—figure supplement 1*. The online version of this article includes the following video and figure supplement(s) for figure 2:

**Figure supplement 1.** Time-lapse imaging of the DTC-Sh1 interface after gonad migration shows a dynamically remodeled Sh1 cell and relatively static DTC.

**Figure 2—video 1.** The DTC stretches while Sh1 grows dynamically to elaborate.

https://elifesciences.org/articles/56383#fig2video1

need to treadmill over germ cells to counter the distal-to-proximal movement of the underlying germ cells from the distal stem cell zone towards the bend (*Hall et al., 1999*). We found support for this hypothesis from time-lapse imaging of Sh1 and the DTC (*Figure 2—video 1*, *Figure 2E*). While the DTC was sometimes observed to stretch proximally, Sh1 almost always (n = 12/13) appeared to rapidly remodel its edges, moving both towards and away from the DTC on a scale of minutes (*Figure 2—figure supplement 1*). These observations suggest Sh1 is a more active cell than the DTC, which changes shape on a slower time scale (*Wong et al., 2013*). Time course analysis of the same cells revealed that Sh1 persisted in the distal zone well into adulthood (n = 18/18 animals 73 hr post L1 arrest, *Figure 2B* bottom). Taken together, these results indicate that cellular extensions from the Sh1 cells interdigitate with the processes from the DTC as the DTC elaborates and are then present in this interlaced pattern during adulthood.

## Nearly every germ cell contacts the DTC, Sh1, or Both

Because Sh1 extensions are interlaced among DTC processes, we postulated that they likely interact with the actively dividing germ cells in the progenitor zone, including perhaps even germ stem cells in the DTC niche. We hypothesized that progenitor zone cells reside in four different anatomical compartments defined by their somatic cell contacts—DTC-associated, Sh1-associated, DTC-Sh1 interface, or not touching either somatic cell. Since the DTC is necessary and sufficient to maintain a mitotic germ cell population (*Austin and Kimble, 1987*; *Kimble and White, 1981*), these contacts could directly inform the regionalization of the mitotic zone into stem cells and mitotic cells that have lost their stemness.

To examine germ cell-somatic cell associations in the progenitor zone, we generated worms with fluorescent cell membrane markers of the germ cells, DTC, and mKate2::INX-8 marking Sh1. We classified the anatomical positions of germ cells in distal gonads (*Figure 3A*). We focused only on the superficial portion (~10 µm deep) of each gonad arm because resolution and brightness diminish when imaging deeper. Viewed at 1000x magnification, we observed an average of 70 undifferentiated germ cells in the superficial distal gonad (n = 26 distal gonad arms). The DTC alone contacted an average of 17 germ cells in this region (*Figure 3B*). Sh1 alone contacted an average of 40 germ cells (*Figure 3B*). An average of 10 germ cells were found along the DTC-Sh1 interface (*Figure 3B*). Finally, very few germ cells (fewer than two germ cells per animal) lacked close proximity to either somatic cell (*Figure 3B*). Thus, Sh1 is intimately associated with the majority of germ cells at the distal end of the gonad, and practically all germ cells in the progenitor zone contact either the DTC, Sh1, or both (*Figure 3B*).

## A disproportionate number of divisions occur at the DTC-Sh1 interface

Germ cell division dynamics are nonuniform across the progenitor zone, with a peak in the incidence of divisions between 20–40 µm from the distal tip (*Hansen et al., 2004*; *Maciejowski et al., 2006*). Notably, this is where the DTC-Sh1 interface falls (*Figure 3C and D*), so we next investigated germ cell divisions in the context of their somatic cell compartments.

We used live in-animal imaging to measure both the distance from the distal tip of the gonad and the somatic cell contacts of 82 cell divisions in 40 young adult animals in a worm strain in which the DTC, Sh1, germ cell membranes, and germ cell nuclear histones were fluorescently highlighted. The absolute positions of DTC-associated, Sh1-associated, and DTC-Sh1 interface cell divisions overlap

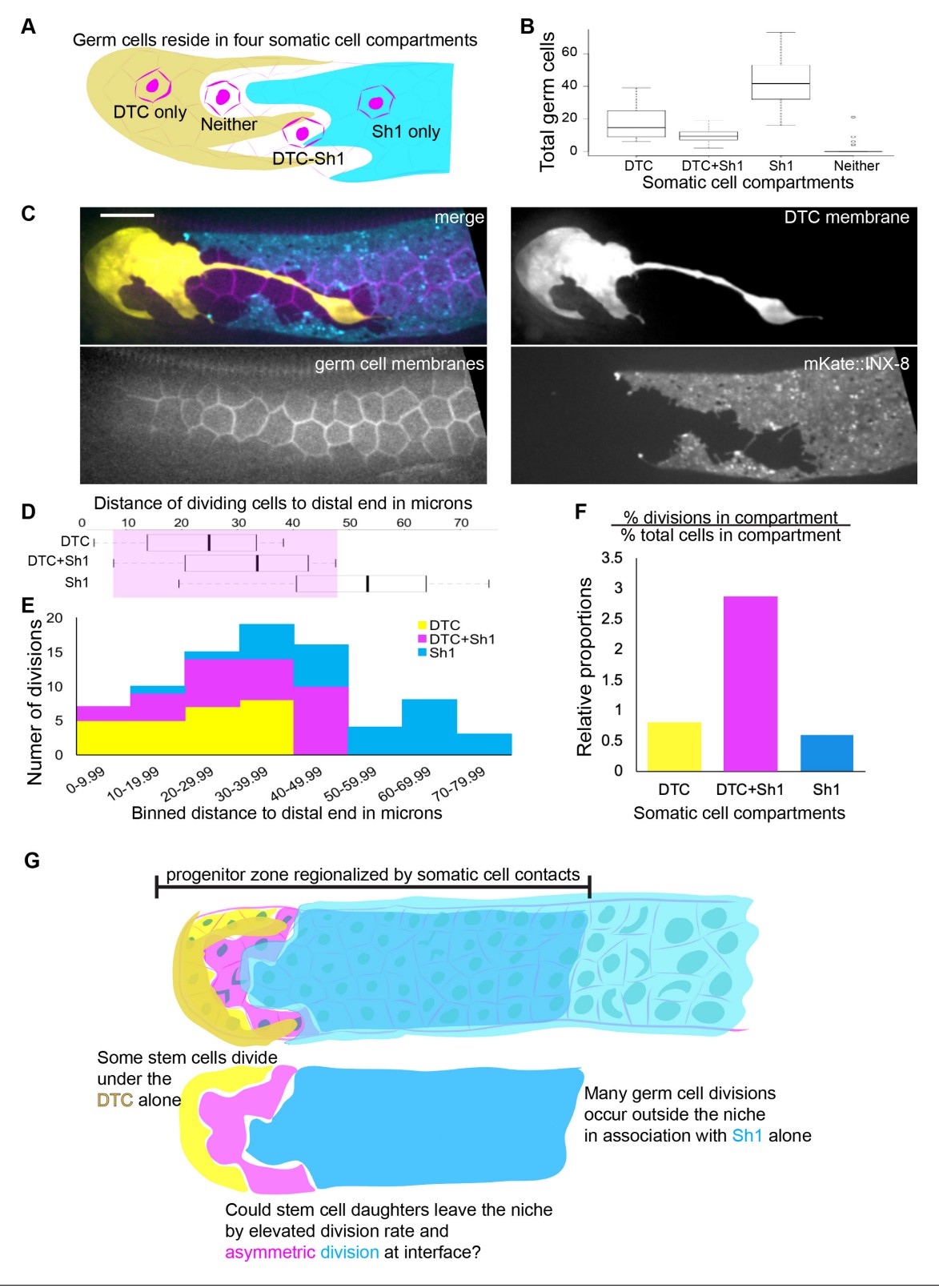

**Figure 3.** Most germ cells contact one or both somatic cells, and divisions are enriched at the DTC-Sh1 interface. (**A**) Schematic depicting germ cells in four somatic cell compartments defined by contact with the DTC, Sh1, DTC-Sh1, or neither. (**B**) Box and whisker plot showing the number of germ cells in each somatic compartment in 26 one-day adult animals (n = 1843 germ cells counted *Figure 3—source data 1.*). Very few germ cells contact neither somatic cell, while a minority contact both the DTC and Sh1. (**C**) Example micrograph showing that all germ cell membranes (magenta, *mex-5p::GFP::*

*Figure 3 continued on next page*

Figure 3 continued

$PLC^{\delta PH}$) in the distal gonad contact the soma (DTC: yellow, *lag-2p::mTag:BFP::PLC^{\delta PH}* and/or Sh1: cyan, *mKate2::inx8*) with no germ cell bare region. Scale bar 10 μm, and images are to scale with distance axes in panels D and E. (D) Distance from the distal tip of n = 25 DTC-associated divisions, n = 29 DTC-Sh1 divisions, and n = 28 Sh1-associated divisions in 38 adult animals with germ cell nuclei (*mex-5p::H2B::mCherry*) and membranes (*mex-5p::GFP:: PLC^{\delta PH}*), DTC membrane (*lag-2p::mTag:BFP::PLC^{\delta PH}*) and Sh1 membrane (*mKate2::inx8*) labeled *Figure 3—source data 2.* No divisions of a 'bare' germ cell were observed. Shaded box shows range of distances in which the DTC and Sh1 were both sometimes observed. (E) The same 82 germ cell divisions as in D, plotted by binned distance from the distal tip, and colored by somatic cell contact with DTC (yellow), DTC-Sh1 (magenta), and Sh1 (cyan). (F) Relative proportions of germ cell division to total germ cells among 25 time-lapsed cell divisions in nine animals *Figure 3—source data 3.* These data are concordant with proportions for all measured divisions in each compartment over all counted cells in each compartment *Figure 3—source data 2.* The percentage of germ cell divisions in contact with the DTC alone is proportional to the percentage of all germ cells in contact with the DTC alone (yellow, p=1.0). Divisions in contact with Sh1 alone are slightly underrepresented (cyan p=0.13), likely because some of the germ cells in contact with Sh1 are already in meiotic S phase and therefore are not able to undergo mitosis (*Fox and Schedl, 2015*). Divisions at the interface (pink) are overrepresented threefold relative to the percentage of DTC-Sh1 associated germ cells (prop.test, Bonferroni corrected p=0.0012327, see Materials and methods). (G) Schematic depicting progenitor zone regionalization by somatic cell contacts. The territory in which dividing germ cells are found (the progenitor zone) contains three groups of dividing cells (uncovered germ cells in 3A and 3B were not found in substantial numbers and were not observed dividing). Previous research suggests the cells directly in contact with the DTC are stem cells, but we now can resolve two groups—those contacting the DTC alone (yellow) and DTC-Sh1 interface cells (magenta). The remaining dividing cells are outside the niche and in contact with Sh1 alone (cyan). Based on frequent divisions that segregated one daughter cell under the Sh1 in DTC germ cells that divided at the DTC-Sh1 boundary, we hypothesized that DTC-Sh1 interface cells might divide asymmetrically and thereby allow stem cell daughters to leave the niche. See also *Figure 3—video 1* and *Figure 3—figure supplement 1*.

The online version of this article includes the following video, source data, and figure supplement(s) for figure 3:

**Source data 1.** Somatic cell contacts of germ cells in superficial portions of 26 young adult hermaphrodite gonads.
**Source data 2.** Somatic cell contacts of 82 dividing cells.
**Source data 3.** Somatic cell contacts of 25 dividing cells in 9 young adult hermaphrodite gonads with germ cell nuclei and membranes labeled.
**Figure supplement 1.** Example of germ cell divisions scored for *Figure 3D–F*.
**Figure 3—video 1.** Germ cells divide at the DTC-Sh1 interface and segregate between the two.
https://elifesciences.org/articles/56383#fig3video1

---

in the distal 10–50 μm of the gonad (*Figure 3D*, shaded box), reflecting the variable, uneven, inter-calating boundary between these somatic cells. While a minority of germ cells lie on the DTC-Sh1 interface (fewer than 15% of all cells, *Figure 3B*), these cells made roughly one third of total germ cell divisions observed (n = 29/82, *Figure 3E*, pink). Another third occurred in DTC-associated germ cells (n = 25/82, *Figure 3E*, yellow), and a third occurred in Sh1-associated germ cells (n = 28/82, *Figure 3E*, cyan). The distance distribution of these divisions roughly recapitulates the mitotic index peak pattern previously observed between ~20–40 μm from the distal end (*Hansen et al., 2004*; *Maciejowski et al., 2006*), and the extent of Notch target transcription among distal germ cells, with no transcripts being made beyond 40 μm from the distal end (*Lee et al., 2019*; *Lee et al., 2016*). This correspondence suggests that the stem cell zone extends to the DTC-Sh1 interface, and that combined DTC and Sh1 contacts promote cell division.

We next collected a dataset to explicitly test the proportionality of divisions across compartments. We analyzed ten time-lapse movies for which we could ascribe germ cell-somatic cell contacts for 25 cell divisions over about half an hour each (see example in *Figure 3—figure supplement 1*). We found that the number of divisions occurring at the DTC-Sh1 interface is dispro-portionate to the number of cells in that compartment (*Figure 3F* pink, prop.test, Bonferroni corrected p=0.0012327, see Materials and methods). In contrast, the shares of divisions under the DTC alone and Sh1 alone are proportional if not somewhat underrepresented (*Figure 3F*, corrected p values for DTC p=1.0, yellow and Sh1 p=0.13, cyan). We concluded that germ cell divisions are over-represented on the DTC-Sh1 boundary.

## Sh1 actively segregates germ cell daughters at the DTC-Sh1 interface

During our live-cell imaging, we noted that germ cell divisions at the DTC-Sh1 interface appeared to be polarized between the two somatic cells, with each condensation of chromatin noticeably closer to either the DTC or Sh1 cell (*Figure 3—video 1*, and *Figure 3—figure supplement 1*). We were intrigued by the possibility that external somatic cell contacts might provide a mechanism that dic-tates stem cell retention vs. exit from the niche (see schematic *Figure 3G*). The Notch ligand LAG-2 is localized to the DTC membrane (*Gordon et al., 2019*; *Henderson et al., 1994*), and losing active

Notch signaling is necessary for germ cells to differentiate (*Kershner et al., 2014*; *Maciejowski et al., 2006*). We next asked how germ cells could escape this contact-mediated signal from the DTC niche and become able to differentiate. We hypothesized that these cells might either lose DTC contact and gain Sh1 associations by moving, initiate an exclusive association with Sh1 at birth via polarized cell division, or both. We examined how germ cells form associations with Sh1 by time-lapse imaging a strain that expressed GFP::INX-9 to mark the membrane of Sh1 (it is also expressed in puncta in the DTC), as well as a DTC membrane marker and germ cell nuclear histone marker (*Figure 4*). We focused on cell divisions associated with the DTC niche, including those at the DTC-Sh1 interface (that is, not cells dividing under Sh1 alone).

We collected a total of 48 time-lapse movies of 15 min or longer in which a total of 126 germ cells in contact with the DTC divided. About half of these divisions (69/126) occurred at the distal edge of Sh1, at the DTC-Sh1 interface. Because some of these divisions did not resolve during the time-lapse interval or sunk out of the plane of focus, 64 were scored further (*Supplementary file 1*). All 64 divisions ended with at least one daughter cell directly beside or under the Sh1 cell. In most of these cases (56/64) one daughter remained under the DTC or at the interface and the other ended up under or beside Sh1, suggesting a polarized division. In the rare instances in which both daughter cells stayed at the DTC-Sh1 interface (8/64 divisions), the divisions still appeared polarized between the DTC and Sh1, but irregular cellular projections from these cells extended the DTC-Sh1 interface beyond the division plane such that both cells remained at the interface. Intriguingly, in 26/64 cases, Sh1 actively grew during the time-lapse interval to cover the closest daughter or grew into the cytokinetic furrow between the daughters to enwrap the closest daughter (see example in *Figure 4*). These dividing interface cells were the only cells (out of 468 total interface cells examined) that changed their germ cell-somatic cell contacts during observation. Similarly, out of the 55 scored cell divisions that occurred in contact with the DTC alone (*Supplementary file 1*), none of the daughters appeared to leave the niche, so division alone is not sufficient to displace a daughter cell out of the niche. These results offer compelling evidence that DTC-Sh1 interface germ cell division, often followed by Sh1 growth over one daughter cell, is the primary mechanism of stem cell niche exit of germ cells.

## Germ stem cell and differentiation markers segregate at DTC-Sh1 interface

If germ cells leave the stem cell niche at the DTC-Sh1 interface, then we would expect DTC-associated germ cells to express a stemness marker and Sh1-associated germ cells to express a marker of meiotic differentiation, and for the boundary between these populations to coincide with the DTC-Sh1 interface. We used a *sgyl-1p::H2B::gfp* transgene (*Kershner et al., 2014*) as a direct target of active Notch signaling in the germ stem cells (*Figure 5A*). The promoter comes from the gene *sygl-1*, which acts directly downstream of active Notch signaling (*Kershner et al., 2014*) to promote stemness in the presence of FBF proteins (*Shin et al., 2017*). We use a *gld-1::gfp* transgene (*ozIs5*; *Brenner and Schedl, 2016*; *Schumacher et al., 2005*; *Seidel et al., 2018*) as a marker of differentiation (*Figure 5A*). The *gld-1* gene encodes an RNA binding protein that is not abundant in distal mitotic cells (*Jones et al., 1996*); it is post-transcriptionally repressed by *sygl-1* and FBF in the stem cell zone (*Brenner and Schedl, 2016*; *Crittenden et al., 2002*; *Shin et al., 2017*). GLD-1 RNA-binding protein promotes meiotic entry (*Biedermann et al., 2009*; *Jungkamp et al., 2011*; *Kadyk and Kimble, 1998*). We combined these transgenes with markers of Sh1 and the DTC from this study (*Figure 5A*).

The *sygl-1p::H2B::gfp* transgene is prone to silencing, and is quite variable among animals. However, we observed 31/36 adult animals had the region of brightest GFP expression limited to the cells that fall distal to the DTC-Sh1 interface, inclusive of interface cells *Figure 5—source data 1*. We found that the long DTC processes in older adults were not correlated with GFP expression in the underlying germ cells (*Figure 5B*); this is consistent with previous reports showing that *sygl-1p::*H2B::GFP expression localizes to germ line regions with dense, short DTC processes (*Linden et al., 2017*). We now know that the distal-most density of DTC processes is directly bordered by Sh1. This finding is consistent with cells under the DTC and at the DTC-Sh1 interface having a molecular signature of stemness, and cells outside of this domain under Sh1 alone losing that signature.

The GLD-1 protein and GLD-1::GFP expressed by a transgene have apparent 'steps' in intensity (*Brenner and Schedl, 2016*; *Seidel et al., 2018*), which were recently shown to reflect the folding of

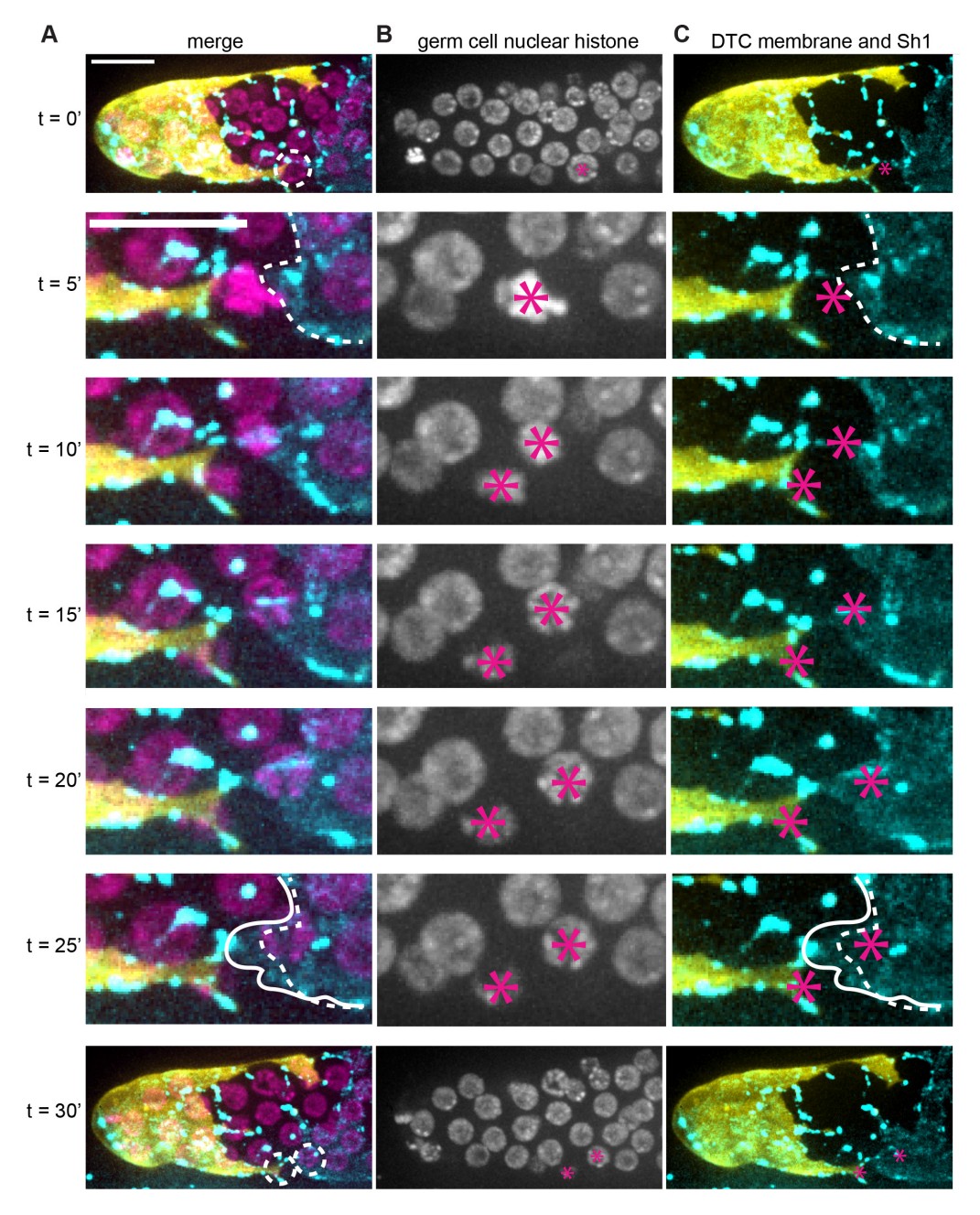

**Figure 4.** Sh1 separates daughter cells that divide at interface. (**A**) DTC membrane (yellow, *lag-2p::mTag:BFP::PLC^{δPH}*), germ cell nuclei (magenta, *mex-5p::H2B::mCherry*), and Sh1 membrane (cyan, *GFP::inx9*, also expressed in puncta on DTC processes). Dashed circles in top and bottom rows marks focal germ cells enlarged in intervening timepoints; dashed outline in second and sixth rows denote starting position of Sh1 membrane. Solid outline in sixth row indicates Sh1 membrane that has extended beyond its starting position to cover dividing cell. (**B**) Germ cell nuclei (magenta, *mex-5p::H2B::mCherry*) channel only. Asterisks mark nuclei of focal cell(s). (**C**) DTC membrane (yellow, *lag-2p::mTag:BFP::PLC^{δPH}*) and Sh1 membrane (cyan, *GFP::inx9*) channels with germ cell asterisks from (**B**) and annotations from (**A**). 1000x magnification. Scale bar 10 μm. See also *Figure 4—video 1*.

The online version of this article includes the following video for figure 4:

**Figure 4—video 1.** Sh1 separates daughter cells at DTC-Sh1 interface.

https://elifesciences.org/articles/56383#fig4video1

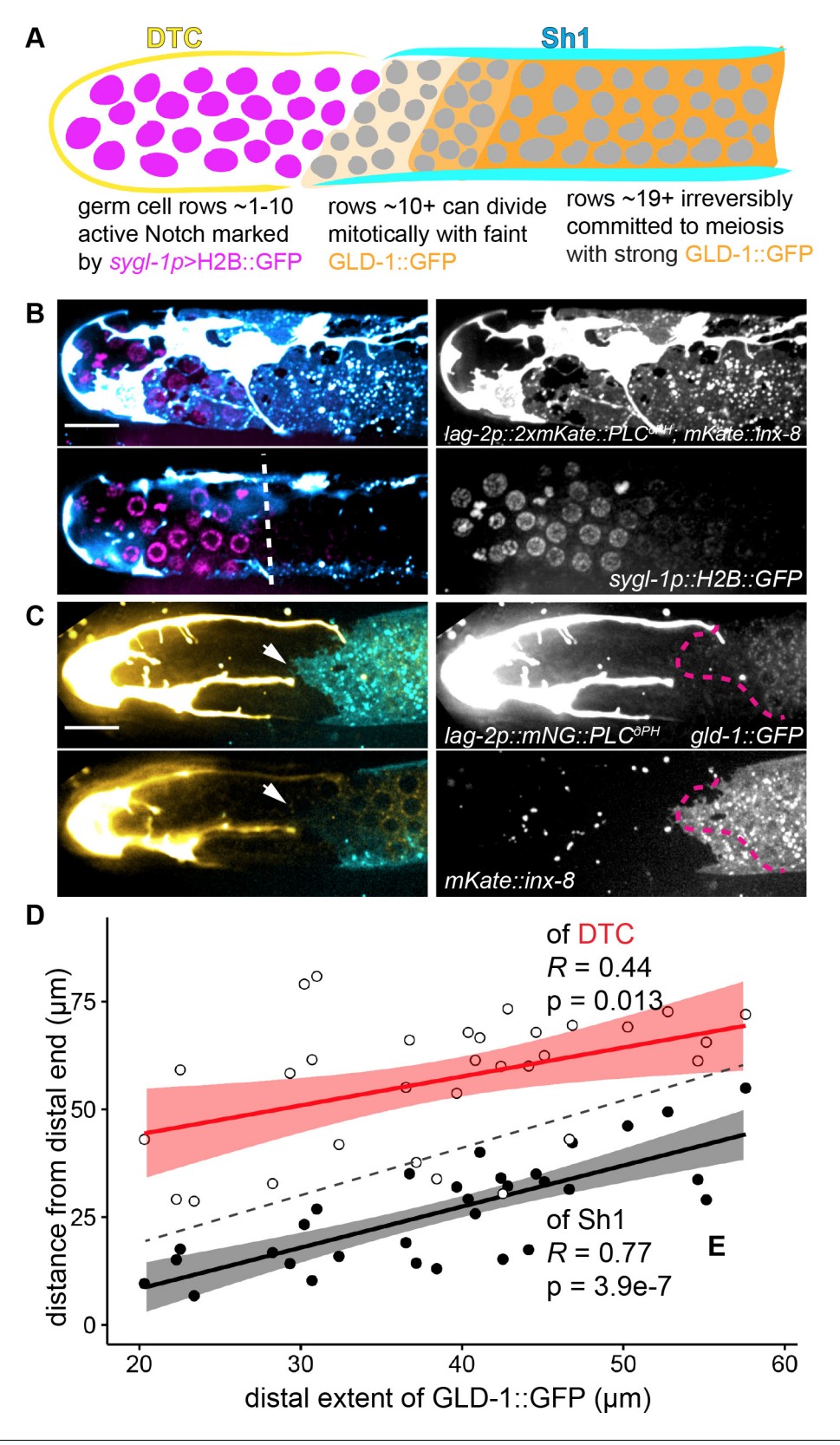

**Figure 5.** Genetic markers of germ cell fate segregate across DTC-Sh1 interface. (**A**) Schematic depicting cross section of *C. elegans* gonad with regionalization of germ cell fate markers. A *sgyl-1p::H2B::gfp* transgene

*Figure 5 continued*

(*Kershner et al., 2014*) is a direct target of active Notch signaling in germ stem cells (magenta). *gld-1::gfp* transgene (*ozIs5*) is post-transcriptionally repressed in the stem cell zone (*Brenner and Schedl, 2016*) and its protein is found at low baseline levels there;it becomes de-repressed and protein accumulates in mitotic germ cells farther from the DTC (light and medium orange shading), and is highly expressed in germ cells committed to meiosis (dark orange shading). Previously reported regions of expression roughly correlate with our observed locations of the DTC (yellow) and Sh1 (cyan) cells. (B) Expression of *sgyl-1p::H2B::gfp* (magenta) with *lag-2p::2xmKate::PLC$^{\delta PH}$* (cyan/white) marking DTC and *mKate::inx-8(qy78)* (cyan) marking Sh1. Surface z-projection (top left) and cross section (bottom left), with white dashed line marking the boundary of brightest *sgyl-1p::H2B:: gfp* expression. Grayscale images at right. (C) Expression of *gld-1::GFP(ozIs5)* (yellow) with *lag-2p::mNeonGreen:: PLC$^{\delta PH}$* (yellow/white) marking the DTC and Sh1 (cyan, *mKate2::inx-8*) marking Sh1. Surface z-projection (top left) and cross section (bottom left), with white arrows indicating the distalmost edge of Sh1. Grayscale images at right, with magenta dashed line showing distalmost boundary of detectable GLD-1::GFP. Note correlation between Sh1 and GLD-1::GFP. (D) Plot of positions in 31 animals of proximal extent of DTC (empty circles, regression in red) or distal extent of Sh1 (filled black dots, regression in black) vs. distal extent of GLD-1::GFP *Figure 5—source data 2.* R and p values given on graph; confidence intervals shaded around regression lines. Dashed line indicates x = y; data points above this line have a distal boundary of GLD-1::GFP expression that is closer to the distal end than the somatic cell, while data points below the line have a distal boundary of GLD-1::GFP that is farther from the distal end than the somatic cell.

The online version of this article includes the following source data for figure 5:

**Source data 1.** sygl-1p:: H2B::GFP::sygl-1 3′UTR expression relative to somatic cells.
**Source data 2.** gld-1::GFP expression and somatic cell boundaries.

---

the germ cell syncytium, along which there is a smooth increase in GLD-1 intensity along the plane of the rachis from distal (low/absent) to proximal (high) (*Seidel et al., 2018*). In 31 young adult animals, we saw the distalmost GLD-1::GFP signal correlated with the distal edge of Sh1 (*Figure 5C and D*), supporting our hypothesis that distal Sh1-associated cells are in the early stages of differentiation. In fact, they seem to correspond with the position in the gonad at which germ cells are irreversibly committed to meiosis (*Brenner and Schedl, 2016*; *Cinquin et al., 2010*)—proximal to the densest DTC processes, but distal to the transition zone in which nuclear morphology is clearly meiotic. In most animals, processes of Sh1 extended just distal beyond the distal extent of GLD-1::GFP signal, meaning we observed GLD-1::GFP in only Sh1-associated cells (the data fall beneath the x = y line shown in gray dashes *Figure 5D*). On the other hand, while the DTC's proximal extent is more weakly correlated with the distal extent of GLD-1::GFP (*Figure 5D*), an anticorrelation that was previously observed (*Byrd et al., 2014*), long DTC processes typically run deep into the region of active GLD-1::GFP (the data consistently fall above the x = y line shown in gray dashes). Taken together, the patterns of *sygl-1p*::H2B::GFP and GLD-1::GFP expression strongly support cell fate asymmetry across the DTC-Sh1 boundary that segregates stem cells from their differentiating progeny.

## Germ cell spindles orient to the somatic cells

Given that germ cell fates segregate across the DTC-Sh1 boundary, we next asked how germ cells traverse the boundary. Because most (56/64 interface divisions, see above) of the DTC-Sh1 interface divisions we observed segregated their daughters asymmetrically (as in *Figure 4*), we hypothesized that the somatic cells orient the germ cell divisions at the interface. Oriented cell division is a common feature of stem cells (*Morrison and Kimble, 2006*) but has not been observed in *C. elegans* germ cells, which are instead thought to divide symmetrically and without orientation (*Crittenden et al., 2006*; *Crittenden et al., 2003*; *Morrison and Kimble, 2006*).

To resolve whether germ cells made oriented divisions at the DTC-Sh1 boundary, we performed time-lapse imaging and recorded 97 mitoses in a strain expressing beta-tubulin::GFP (*Galy et al., 2003*; *Gerhold et al., 2015*; *Strome et al., 2001*), a marker of the mitotic spindle. The 29 germ cell divisions at the DTC-Sh1 interface had spindles that were more aligned to the shortest line connecting the DTC to Sh1 than to the distal-proximal axis of the gonad (the null hypothesis, *Figure 6B–E*, one sided Wilcoxon rank sum test W = 141, p-value=2.537e-06). When cells divided on the DTC-Sh1 boundary, they oriented their spindles toward these two cells and divided asymmetrically (with one

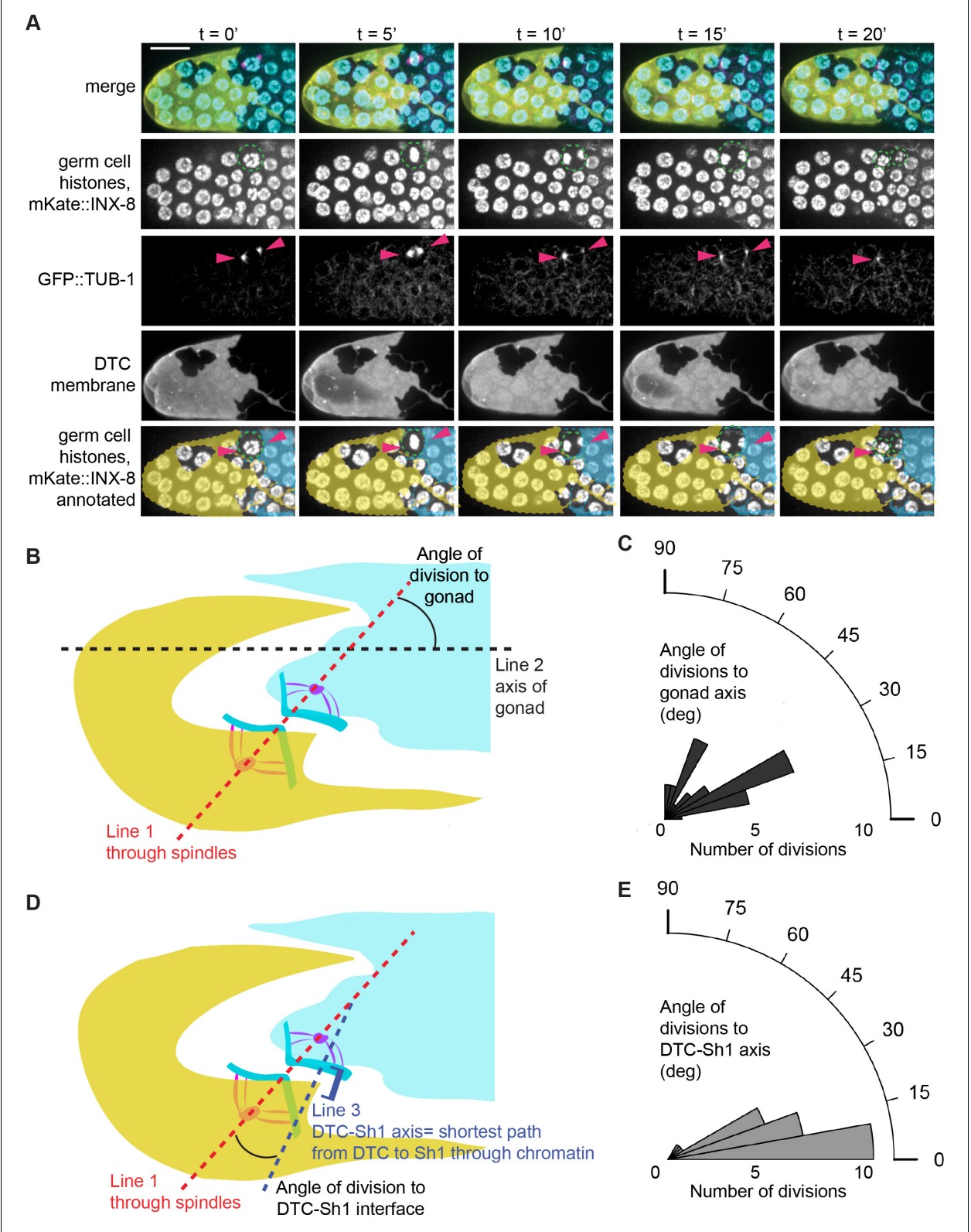

**Figure 6.** Tubulin spindles polarize to the DTC and Sh1 as germ cells divide at the interface. (A) Stills from a time-lapse movie (*Figure 6—video 1*) of a germ cell dividing (dashed outline in second row) at the DTC-Sh1 interface in a young adult animal. DTC (yellow, *lag-2p::mTag:BFP::PLC*$^{\delta PH}$), Sh1 (cyan, *mKate2::inx8*), germ nuclei (cyan, *mex-5p::H2B::GFP*), and tubulin (pink, *β-tub::GFP*). Tubulin spindles are transiently visible (arrowheads in third row). Yellow and cyan overlays in the bottom row mark cell outlines of DTC and Sh1, respectively. Scale bar is 10 µm. (B) Schematic depicting how angle of

*Figure 6 continued on next page*

*Figure 6 continued*

division to gonad was calculated. For each cell dividing at the interface between the DTC and Sh1 (n = 29) *Figure 6—source data 1.* Line one was drawn through the spindles into the neighboring soma. Line two was drawn along the distal-proximal axis of the gonad. The angles between Line one and Line two reflect how aligned the spindles are to the axis of the gonad, and are plotted in (C). (D) Schematic depicting how angle of division to DTC-Sh1 interface was calculated. Line one again was drawn through the spindles and into the soma (same as in B). Line three was drawn through the shortest path between DTC and Sh1 that passes through at least one of the chromatin condensations at anaphase; this is the DTC-Sh1 axis. The angles between Line one and Line three reflects how aligned the spindles are to the DTC-Sh1 axis, and are plotted in (E). Note that every angle was expressed as an acute angle for the sake of consistency and to increase power. A Wilcoxon rank sum test was performed on the angles shown in C and E, and the result shows that they are significantly different from each other with W = 91 and p=1.967e-05, with the division angles being more aligned to the DTC-Sh1 axis. See also *Figure 6—video 1* and *Figure 6—figure supplement 1*.

The online version of this article includes the following video, source data, and figure supplement(s) for figure 6:

**Source data 1.** Division angles of 97 germ cell mitoses.
**Figure supplement 1.** Germ cell divisions under the DTC and Sh1, like germ cells dividing between DTC and Sh1, are not oriented towards the gonad axis.
**Figure 6—video 1.** Tubulin spindles polarize between the DTC and Sh1 as a germ cell divides at the interface.
https://elifesciences.org/articles/56383#fig6video1

---

daughter associated with the DTC and the other toward Sh1, *Figure 6A*). Spindle orientation was observed across the DTC-Sh1 interface prior to division, which is consistent with a model in which the DTC-Sh1 interface dictates the asymmetric division (as opposed to the completed cell division later triggering remodeling at the DTC-Sh1 interface).

The absence of orientation with respect to the distal-proximal gonad axis was also observed for cells that divided under the DTC alone, and cells that divided under Sh1 alone (each Wilcoxon test for division angle to gonad between pairs of the three compartments has a p>0.1, *Figure 6—figure supplement 1*). The absence of orientated division to the gonad axis has been construed by others as evidence against oriented divisions of adult *C. elegans* germ cells (*Crittenden et al., 2006*; *Morrison and Kimble, 2006*). Our results, however, offer compelling evidence that the geometrically complex and variable DTC-Sh1 interface has obscured a clear view of strongly oriented divisions along this cryptic anatomical boundary.

## Distal sheath extension requires the F-actin regulators Arp2/3 and Cofilin

Spindle orientation is a key feature of germ cell division at the DTC-Sh1 interface; the other key feature is the 'grabbing' of one daughter at cytokinesis by growth of the Sh1 cell. We hypothesized that the dynamic reaching of Sh1 may be actin-dependent, and that disrupting the actin cytoskeleton during adulthood may cause Sh1 to fail to maintain distal membrane extensions over the proliferating germ cells as their progeny are displaced proximally.

To test if actin remodeling was required for Sh1 to maintain its distal position near the DTC, we devised a developmentally timed RNAi treatment to knock down two crucial F-actin regulators, a branched actin nucleator Arp2/3 subunit encoded by *arx-2* and the *C. elegans* ortholog encoding the F-actin disassembly protein cofilin (*unc-60*). Both *arx-2* (*Roh-Johnson et al., 2012*; *Sawa et al., 2003*) and *unc-60* (*Ono et al., 2003*) are essential genes required for embryonic development. To avoid developmental defects caused by genetic loss or early RNAi knockdown (*Ono et al., 2008*), we fed early L4 larval animals bacteria expressing RNAi targeting *arx-2* (*Figure 7—figure supplement 1*) and *unc-60*. Consistent with our hypothesis, in adults two days after RNAi exposure there was a gap between the DTC and Sh1 (*Figure 7A*), and Sh1 took on a stringy, fragmented form at its distal end while the DTC structure was not similarly perturbed. While worms fed empty RNAi vector had an average distal position of Sh1 that fell 44 μm from the distal end of the gonad, *unc-60* RNAi-treated worms had an average distal position of 84 μm (*Figure 7B*). A linear mixed model revealed that the positions in the RNAi treated worms are different from the control worms (linear mixed model: $F_{1,62}$ = 40.00675 Tukey post-hoc test p<0.0001). Further supporting the notion that F-actin-mediated dynamic membrane extensions promote maintenance of Sh1 position, we found that an endogenously tagged Arp2/3 subunit encoded by *arx-2::gfp* (*Kelley et al., 2019*; *Zhu et al., 2016*) localized dynamically to the edge of the Sh1 cell during membrane extensions over the dividing daughter cell closest to the Sh1 cell (*Figure 7—video 1*, *Figure 7—figure supplement 1D* and

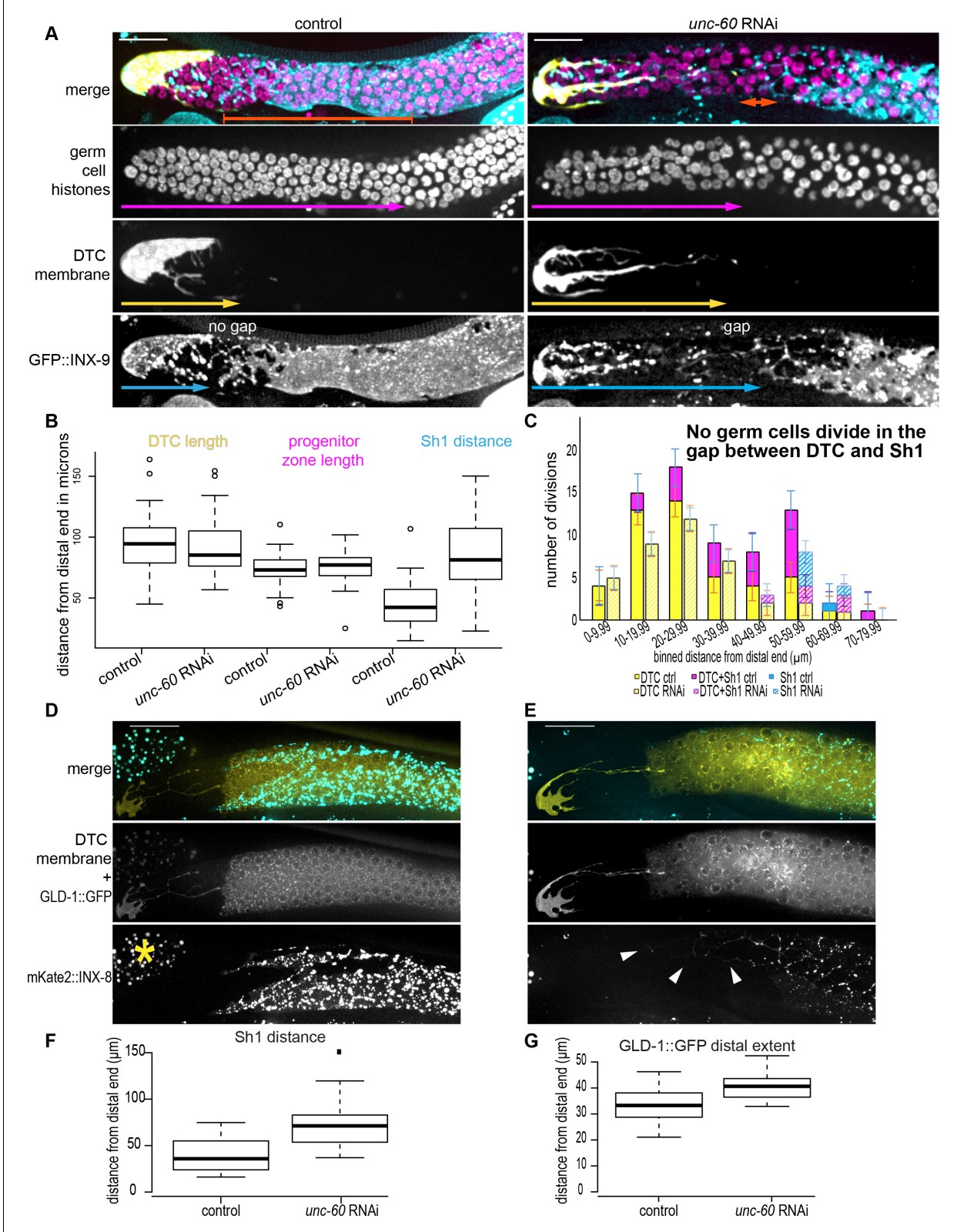

**Figure 7.** The distal sheath maintains proximity to the DTC via cofilin-dependent growth, and loss of proximity correlates with diminished germ cell proliferation and regulated niche exit. (**A**) The DTC-Sh1 interface (DTC in yellow, *lag-2p::mTag:BFP::PLC*$^{\delta PH}$; Sh1 in cyan, *GFP::inx-9* bottom) seen in in the control (left) is disrupted by RNAi (right) against gene encoding worm ortholog of actin binding protein cofilin (*unc-60*). Germ cell nuclei (in magenta, *mex-5p::H2B::mCherry*) can be observed in the undifferentiated (distal, smaller) and meiotic (crescent and larger nuclei) states. Crescent

*Figure 7 continued on next page*

*Figure 7 continued*

shaped nuclei mark the meiotic transition zone. (B). Neither the DTC length (yellow arrows in A) nor the length of the progenitor zone (magenta arrows in A) are affected by RNAi treatment, while the distal boundary of Sh1 (cyan arrows in A) is perturbed 0: $F_{1,62}$ = 40.00675 Tukey post-hoc test p<0.0001. In all graphs, boxes show median and second and third quartiles, while whiskers show minimum and maximum values. (C) Binned distances from the distal tip of germ cell divisions in three somatic cell compartments: DTC-alone (yellow), DTC-Sh1 (magenta), Sh1-alone (cyan) for control (solid fill) and *unc-60* RNAi-treated (hashed) worms 48 hr after exposure beginning in the L3/L4 stage *Figure 7—source data 1.* Note that this is ~1 day later than the animals analyzed in *Figure 3E*, and the DTCs are substantially more elongated by this stage. Control n = 31 worms, *unc-60* RNAi n = 35 worms. Representative images of (D) control and (E) *unc-60* RNAi-treated worms expressing DTC membrane GFP, GLD-1::GFP transgene, and mKate2::INX-8 to mark Sh1. Yellow asterisk marks autofluorescent gut granules in neighboring tissue. Arrowheads mark distal strings of retracted Sh1. (F) Boxplots showing distance from distal end of Sh1 in control and *unc-60* RNAi-treated worms *Figure 7—source data 2.* Welch two-sample t-test p<0.001. Note concordant results to those shown in (B). (G) Boxplots showing distance from distal end of GLD-1::GFP expression *Figure 7—source data 2.* Welch two-sample t-test p<0.001.

The online version of this article includes the following video, source data, and figure supplement(s) for figure 7:

**Source data 1.** Gonad feature measurements, positions and somatic cell contacts of dividing cells in control and RNAi-treated animals.
**Source data 2.** Position of Sh1 and GLD-1::GFP boundaries in control and RNAi-treated animals.
**Figure supplement 1.** RNAi against *arx-2* causes the Sh1 distal boundary to be pushed back.
**Figure supplement 1—source data 1.** Positions of Sh1 boundary and transition zone in control and RNAi-treated animals.
**Figure 7—video 1.** As germ cells divide at the DTC-Sh1 interface, Sh1 uses ARX-2::GFP to grow distally and segregate germ cells.
https://elifesciences.org/articles/56383#fig7video1

*Figure 7—figure supplement 1E*, and n = 10/11 time-lapse recordings of interface divisions). We conclude that continuous and sporadic distal sheath growth requires actin remodeling and is required to maintain the Sh1 interface with the DTC.

## Distal Sh1 has a Pro-proliferative effect

The Sh1 cells are important for larval germ line mitotic proliferation (*Killian and Hubbard, 2005*; *Killian and Hubbard, 2004*; *Pepper et al., 2003*), but were thought not to interact with proliferating germ cells in the adult. Given that our results suggested that the DTC-Sh1 interface promoted proliferation, we next assessed the effect of Sh1 and the underlying germ cells by perturbing its position with *unc-60* RNAi. The undifferentiated germ cells at the distal end of the gonad in the progenitor zone are sometimes called 'mitotic' germ cells because of their cell fate (they are not in meiosis, as differentiated germ cells are). Mitotic germ cell nuclei are smaller and less bright by histone fluorescence than meiotic germ cell nuclei, and reside distal to the crescent-shaped nuclei of the meiotic 'transition zone' (*Hubbard, 2007*; *Figure 7A* magenta arrows). However, the mitotic fate must not be conflated with active mitotic events; germ cells can fail to proliferate but also remain undifferentiated (or 'mitotic') (*Seidel and Kimble, 2015*). We hypothesized that if distal Sh1 is pro-proliferative (due to its association with most observed cell divisions, *Figure 3*), its mispositioning by RNAi would cause fewer active divisions in the progenitor zone. Indeed, the average number of actively dividing cells in the progenitor zone in age-matched control animals was 2.29, while the average number of dividing cells in *unc-60* RNAi animals was 1.37 at the endpoint 48 hr after RNAi exposure beginning in the early L4 (n = 31 control, n = 35 experimental, linear mixed model: $F_{1, 62}$ = 5.46813, Tukey post-hoc test p<0.05). While just 5/31 control animals lacked active cell division, 14/35 *unc-60* RNAi animals lacked active cell division (chi-squared = 4.5694, p<0.05). The lengths of the progenitor zones were not different between the *unc-60* RNAi-treated animals and control. However, because Sh1 was shifted so profoundly (*Figure 7B*), the amount of the mitotic germ line covered by Sh1 was dramatically reduced (linear mixed model: $F_{1, 62}$ = 42.566, Tukey post-hoc test p<0.005). While the control animals had about 29 µm of undifferentiated germ cells in contact with Sh1, after treatment with *unc-60* RNAi on average no undifferentiated germ cells contact Sh1. RNAi against *unc-60* can cause germ cell defects under certain exposure conditions (*Ono et al., 2008*). However, the RNAi-treated animals still had normally sized progenitor zones, normally elaborated DTCs (*Figure 7B*), and a number of visibly dividing germ cells.

We assessed the positions and somatic cell contacts of these dividing cells as in *Figure 3E*. The numbers and positions of actively dividing germ cells in contact with the DTC alone are similar in the control and *unc-60* RNAi-treated animals (*Figure 7C*, yellow solid and hashed), indicating that the RNAi treatment did not dramatically affect divisions of germ cells in the DTC plexus. Among the

animals with active divisions (ignoring the 40% of RNAi-treated animals that lack dividing cells entirely), the control (n = 46/26 or 1.77 DTC divisions/animal) and treatment (n = 38/21 or 1.81 DTC divisions/animal) groups had roughly equivalent average numbers of DTC-associated divisions. However, the number of DTC-Sh1 interface divisions was dramatically different between control (n = 23/26 or 0.88 DTC-Sh1 divisions/animal) and treatment (n = 5/21 or 0.24 DTC-Sh1 divisions/animal), since the interface was greatly reduced. We never observed a germ cell dividing without contact to a somatic cell.

Finally, we examined GLD-1::GFP expression in animals after *unc-60* RNAi treatment (*Figure 7D–G*). In this experiment, we again observed a stringy Sh1 phenotype (*Figure 7E*, compare to control in 7D) and a robust proximal displacement of the distal Sh1 boundary (*Figure 7F*, compare to Sh1 distance in 7B). Additionally, we observed a small (~8 μm) but significant (Welch two-sample t-test, p<0.001) shift of the distal boundary of GLD-1::GFP expression in the direction that Sh1 shifted. From this we infer that Sh1, while not required for differentiation, helps germ cells differentiate by maintaining its distal position, possibly by competing with the DTC niche for germ cell contact (*Figure 8*). Taken together, these experiments indicate that the dividing population of undifferentiated, Sh1-associated germ cells are nearly eliminated when Sh1 is mispositioned, leading to an overall deficit of germ cell proliferation and regulated exit from the niche, suggesting that Sh1 is an important but overlooked regulator of germ cells in the progenitor zone.

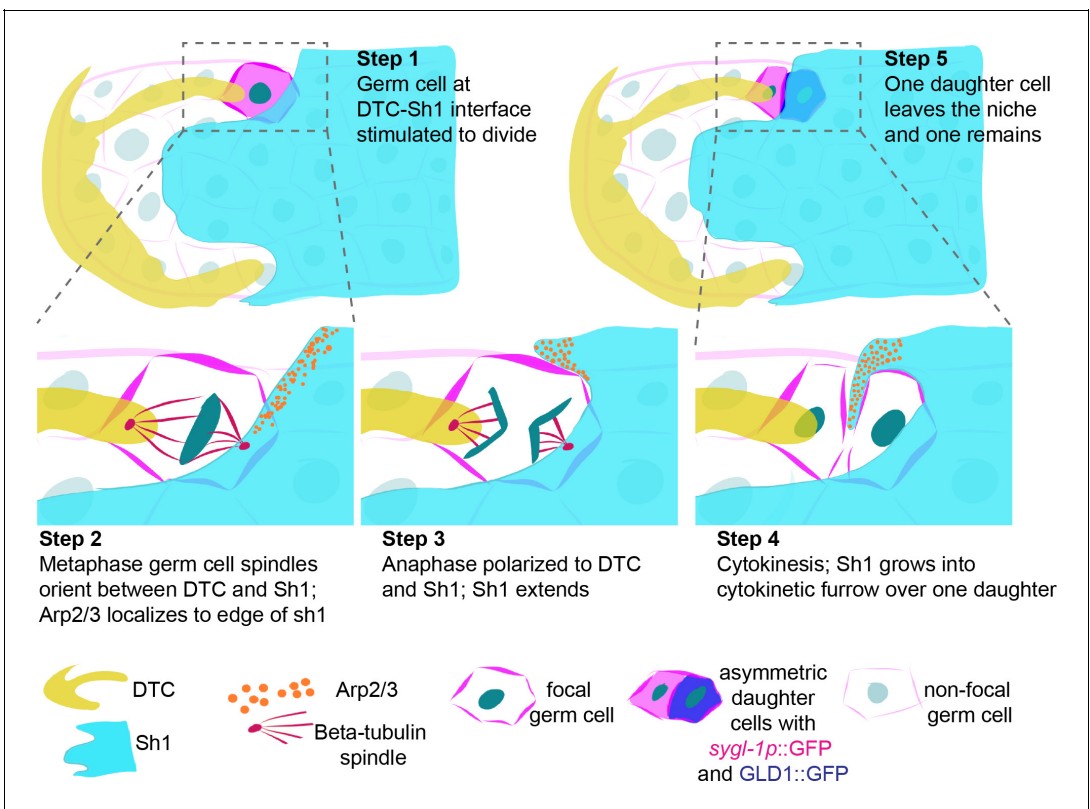

**Figure 8.** Schematic depicting DTC-Sh1 interface germ cell division. (**A**) DTC (yellow) and Sh1 (cyan) contact germ cell membranes (magenta lines) in the distal gonad. Germ cell nuclei (dark turquoise) are visible during mitosis as round nucleus (Step 1), metaphase plate (Step 2), anaphase (Step 3), cytokinesis (Step 4), and in two segregated daughter cells (Step 5). Tubulin spindles (red, Step two and Step 3) and Arp2/3 (orange, Step 2–4) transiently localize in germ cells and Sh1, respectively. As the germ cell divides in a polarized fashion between the DTC and Sh1, Sh1 grows at its distal edge into the cytokinetic furrow to separate the two daughter cells. From birth, these two daughter cells have asymmetric contacts with the soma, and one has exited the niche.

## Discussion

Stem cells rely on their niches to maintain stemness, but their progeny must leave the niche in order to differentiate. Only by studying stem cells and their associated support cells in vivo can we understand how this balance is regulated. Challenges of in vivo imaging have recently been overcome in the zebrafish larval hematopoietic niche, mammalian epidermal stem cell niche, and mouse intestinal crypt, revealing important behaviors such as 'endothelial cuddling' of stem cells (*Tamplin et al., 2015*), different rates of division across stem cells and their daughters, (*Rompolas et al., 2012*), and different trajectories for stem cells depending on their position within the niche (*Ritsma et al., 2014*). By developing genome-edited alleles to act as in vivo cellular markers in the *C. elegans* gonad, we made the surprising discovery that some cells in the *C. elegans* germ line stem cell niche divide in an asymmetric manner towards a niche-adjacent cell. Germ cells that contact this cell instead of the niche have begun to differentiate. This discovery raises the possibility that facilitated niche exit is part of the repertoire of cellular behaviors that regulate the balance between stem cell renewal and differentiation and a new player to a canonical stem cell niche system.

It was previously thought that a germ line bare region extended over most of the progenitor zone, with the Sh1 cells contacting only differentiating germ cells in the adult. By generating endogenously tagged alleles of somatic gonad-expressed innexins, we have demonstrated that adult Sh1 cells extend much farther distal than was previously understood. They contact putative stem cells in the progenitor zone and grow distally into this zone against the proximal flow of germ cells in an Arp2/3- and cofilin-dependent manner. The sheath is known to interact with and promote proliferation of larval germ cells (*Killian and Hubbard, 2005*; *McCarter et al., 1997*), to interact with differentiating adult germ cells to facilitate exit from meiotic pachytene (*McCarter et al., 1997*), and to phagocytose physiologically apoptotic cells (*Li et al., 2012*). It was not thought to contact the progenitor zone in the adult. Our results demonstrate not only that Sh1 processes extend into the progenitor zone, but also that Sh1 plays an important role regulating adult stem cells and their progeny. A somatic gonad cell directly bordering the DTC is also observed in the filarial nematode *Brugia malayi* (*Landmann et al., 2012*; *Foray et al., 2018*), raising the possibility that this cellular arrangement and function are widespread within the clade. During larval development, Sh1 develops in close contact with the DTC niche, implying an intimate relationship between the niche and its neighboring somatic cells as well.

By performing live-cell imaging of cell divisions at the border of the DTC and Sh1 we revealed that while this population comprises a minority of progenitor zone cells, it is more proliferative than cells under the DTC alone or Sh1 alone. As we observed this excess of DTC-Sh1 interface divisions, we noted that dividing cells polarized their spindles between the DTC and Sh1 and divided asymmetrically between them, often with Sh1 actively extending over the nearest daughter. Such divisions were the only times we observed germ cells lose contact with the DTC to exit the niche—we never saw spontaneous growth of Sh1 over a non-dividing cell, nor displacement of germ cells from the niche by distal-to-proximal pushing produced by other cells dividing entirely under the DTC. Previous lineage tracing of germ stem cells and their progeny as they leave the niche (*Rosu and Cohen-Fix, 2017*) revealed that progenitor zone cells move very slowly, with photoconverted populations being displaced about five germ cell diameters over 8 hr. Coupled with the slow rate of division in this population (about 1% of adult germ cells dividing at any given time point [*Roy et al., 2016*]), we think it is reasonable that our time-lapse imaging captured the full range of movements made by the cells in the progenitor zone. Based on these observations, we propose a simple model (*Figure 8*) for germ cell exit from the stem cell niche: germ cells at the DTC-Sh1 interface divide asymmetrically, with one daughter remaining anchored to the DTC and the other becoming enwrapped by Sh1. Those stem cell daughters that maintain their connections to the DTC would comprise an anchored, *sygl-1p::*H2B::GFP positive, asymmetrically dividing stem cell population, while those associating with Sh1 are set on a path to differentiation and dramatically increase levels of GLD-1::GFP. Our results show that Sh1 is not required for germ cell differentiation—GLD-1::GFP is visible in germ cells in the gap between the DTC and Sh1 when Sh1 is shifted proximally—much as we would expect given that differentiation is still observed in the reduced germ cell pools of sheath-ablated animals (*McCarter et al., 1997* and *Killian and Hubbard, 2005*). Instead, we hypothesize that Sh1 promotes differentiation indirectly through its positional exclusion of the DTC from germ cells born from polarized interface divisions (thus allowing these daughter cells to escape the influence of the DTC, which

maintains stem cell fate, *Figure 8*). Since cells born from polarized divisions that contact Sh1 lack DTC-contact-mediated Notch signaling from birth, they can then derepress meiosis-promoting factors and initiate differentiation.

Our findings also have implications for the organization of the progenitor zone of the *C. elegans* germ line. The three somatic cell-associated compartments—DTC, DTC-Sh1 interface, and Sh1—may host three types of progenitor cell divisions: symmetric-stem cell renewing (under the DTC), asymmetric (at the DTC-Sh1 interface), and symmetric-differentiating (or transit-amplifying, under Sh1 alone), which might be comparable to the 'mixed mode' of stem cell division that acts in the mammalian epidermis (*Doupé et al., 2010*; *Rompolas et al., 2016*; *Yang et al., 2015*). This layer of Sh1 control needs to be studied in the context of other factors that influence germ cell proliferation and their increasingly appreciated regulation by age (*Cinquin et al., 2016*; *Kocsisova et al., 2019*; *Seidel and Kimble, 2015*) and environment (*Aprison and Ruvinsky, 2016*; *Roy et al., 2016*).

Most of what we know about how stem cell progeny leave the niche is understood as the absence of the mechanisms that retain the self-renewing stem cell daughter in the niche, be that oriented division to a basal lamina in the mammalian epidermis (*Poulson and Lechler, 2010*) or to niche cells in the *Drosophila* ovary, (*Casanueva and Ferguson, 2004*) or stochastic displacement in mammalian intestine (*Snippert et al., 2010*; *van der Flier and Clevers, 2009*) and epidermis (*Doupé et al., 2010*). One study that specifically addresses how stem cell progeny leave a niche found they do so by migration in the larval tracheal system of *Drosophila* (*Chen and Krasnow, 2014*), a very different mechanism to the one we observe in this study. Cells outside the niche had previously been known to guide stem cell progeny only after they leave the niche. For example in the larval trachea of *Drosophila*, stem cell progeny follow external cues as they migrate from the niche (*Chen and Krasnow, 2014*), and in adult mammalian neurons, niche-adjacent cells send differentiation cues to stem cell progeny (*Ming and Song, 2011*). Some niche-adjacent cells limit the range of niche cues to restrict the stem cell population, like the *Drosophila* testis cyst cells (*Fairchild et al., 2016*; *Fairchild et al., 2015*). Active processes like Sh1 segregating the nearest daughter may take place in other cases of niche exit that appear stochastic, like the mammalian intestinal niche in which 'border cells' are thought to be passively displaced from the niche (*Ritsma et al., 2014*). Similar niche-infiltrating structures may not have been discovered because of the difficulty of observing thin, membranous cellular structures deep inside tissues in vivo that also may not survive dissection or fixation necessary for observing them by immunostaining or electron microscopy, as in the case of cytonemes (*Kornberg and Roy, 2014*). Given the consequences of failing to maintain stem cells and failing to provide sufficient progeny for homeostatic tissue maintenance, mechanisms that balance niche exit with niche retention may be important but overlooked features of many stem cell niches.

## Materials and methods

### Contact for reagent and resource sharing

Further information and requests for resources and reagents should be directed to and will be fulfilled by the Lead Contact, Kacy L. Gordon (kacy.gordon@unc.edu). Strains containing mNeonGreen can only be distributed to labs with a mNeonGreen license from Allele Biotechnology. Strains sourced from *Caenorhabditis* Genetics Center (CGC) are to be requested directly from CGC. Please see Appendix 1: Key Resources Table for more information.

### Experimental model and subject details

*C. elegans* strains were maintained on standard NGM media at 20°C and fed *E. coli* OP50. For RNAi experiments, animals were fed *E. coli* HT115(DE3) containing the L4440 plasmid with or without dsRNA trigger insert (see RNAi experiments, Materials and methods). All animals scored were hermaphrodites (as males have structurally different gonads) at the ages specified in the text.

### Strains

In strain descriptions, we designate linkage to a promoter with a *p* following the gene name and designate promoter fusions and in-frame fusions with a double semicolon (::). Some integrated strains (*xxIs* designation) may still contain *unc-119(ed4)* mutation and/or the *unc-119* rescue transgene in their genetic background, but these are not listed in the strain description for the sake of

concision, nor are most transgene 3′ UTR sequences. Further details available upon request. Strains are as follows:

PS3460 (*unc-119(ed4)*), NK2569 (*xnSi1(mex-5p::GFP::PLCδ^{PH}::nos-2 3′UTR) II; inx-8(qy78(mKate:: inx-8)) IV; qyIS546(lag-2p::mTagBFP::PLCδ^{PH})*), NK2570 (*naSi2(mex-5p::H2B::mCherry::nos-2 3′UTR) II/xnSi1(mex-5p::GFP::PLCδ^{PH}::nos-2 3′UTR) II; inx-8(qy78(mKate::inx-8)) IV; qyIS546(lag-2p:: mTagBFP::PLCδ^{PH})*), NK2571 (*inx-8(qy78(mKate::inx-8)) IV; (cpIs122(lag-2p::mNeonGreen:: PLCδ^{PH})*)), NK2572 (*inx-9(qy79(GFP::inx-9)) IV; cpIs91(lag-2p::2x mKate2::PLCδ^{PH}::3xHA::tbb-2 3′UTR LoxN) II*), NK2573 (*naSi2(mex-5p::H2B::mCherry::nos-2 3′UTR) II; inx-9(qy79(mKate::inx-9)) IV; qyIS546(lag-2p:: mTagBFP::PLCδ^{PH})*), NK2574 (*naSi2(mex-5p::H2B::mCherry::nos-2 3′UTR) II; inx-8(qy78(mKate::inx-8)) IV; qyIS546(lag-2p::mTagBFP::PLCδ^{PH}); ojIs1 [pie-1p::GFP::tbb-2 + unc-119(+)])*, NK2575 (*naSi2(mex-5p::H2B::mCherry::nos-2 3′UTR) II; inx-8(qy78(mKate::inx-8)) IV; arx-2(cas607[arx-2::gfp knock-in]) V*), NK2576 (*inx-9(ok1502) IV; inx-8(qy102(mKate::inx-8)) IV; cpIs122(lag-2p::mNeonGreen::PLCδ^{PH})*)), NK2324 (*qy23 [ina-1::mNG] III*), KLG004 (*cpIs91(lag-2p::2x mKate2::PLCδ^{PH}::3xHA::tbb-2 3′UTR LoxN) II); qSi26(sygl-1p::H2B::GFP::sygl-1 3UTR) II; (inx-8(qy78(mKate::inx-8)) IV; teIs1(oma-1::GFP) IV*), KLG005 (*cpIs122(lag-2p::mNeonGreen:: PLCδ^{PH})); inx-8(qy78(mKate::inx-8)) IV; ozIs5(gld-1:: GFP)*), KLG006 *tnIs6(lim-7p::GFP); inx-8(qy78(mKate::inx-8) IV; (cpIs122(lag-2p::mNeonGreen:: PLCδ^{PH})*)). All strains with multiple genetic elements were generated for this study by crossing the cited strains with genetic elements created for this study (*nasi2* transgene from **Roy et al., 2018**; *xnSi1* parent from **Chihara and Nance, 2012**; *cpIs91* transgene from **Gordon et al., 2019**, *qSi26* and *teIs1* from **Kershner et al., 2014**, *ozIs5* transgene from **Brenner and Schedl, 2016**; **Schumacher et al., 2005**; **Seidel et al., 2018**, *ojIs1* transgene from **Galy et al., 2003** via (**Gerhold et al., 2015**, *arx-2(cas607)* from **Zhang et al., 2017**, *inx-9(ok1502)* allele from **Consortium TC elegans DM, 2012**).

## Molecular biology

We cloned mTagBFP and a *PLCδ^{PH}* membrane-localization domain into the plasmid containing the DTC-expressing *lag-2* promoter fragment generated in **Linden et al., 2017**, and injected this construct into *unc-119(ed4)* mutants with the *unc-119* rescue plasmid and EcoRI-digested salmon sperm DNA. The resulting extrachromosomal array was integrated by standard gamma irradiation protocol.

Genes *inx-8* and *inx-9* are neighboring duplicate genes on Chromosome IV (**Scheme 1A**). The alleles *inx-8(qy78)* and *inx-8(qy102)* were generated using Cas9-triggered homologous recombination with a self-excising selection cassette (**Dickinson et al., 2015**). The target sequence was 5′-GCATCTTCACTCGGGTTCGA**AGG**-3′ with PAM site shown in bold; the guide RNA was cloned into the *eft-3p::Cas9+sgRNA* expression vector pDD162 using this same primer sequence (excluding the PAM). The PAM is seven nucleotides upstream of the start codon of *inx-8*. Homology arms were amplified by PCR from N2 genomic DNA (5′ homology arm forward primer *inx-8* sequences are 5′-tgtacgactgtaggcaggcaggtag-3′ and reverse primer 5′-tctgc<u>aa</u>ttcgaacccgagtgaagatg-3′ (with mutation in PAM site shown underlined); 3′ homology arm forward primer *inx-8* sequences 5′-ATGTTTTC TGTTCCATTTCTTACCTC-3′ and reverse primer 5′-CTGTACATCTCCACGGCAACCTCCG-3′) and cloned into a plasmid modified from **Dickinson et al., 2015** containing a nine amino acid N-terminal leader sequence (with an ATG codon added during assembly), an mKate2 coding gene, a 15 amino acid linker sequence (with TEV site), and a self-excising selection cassette (SEC) flanked by LoxP sites in an intron of mKate2 (**Scheme 1B**). The repair plasmid was coinjected with the sgRNA+Cas9 plasmid into N2 animals, and genome-edited animals were selected by hygromycin B treatment and phenotypic identification (roller). The selection cassette was excised by heat shock as described in **Dickinson et al., 2015**. This construct was also injected into *inx-9(ok1502)* and fertility verified (**Figure 1—figure supplement 1A**), suggesting it is a functional protein fusion.

The allele *inx-9(qy79)* was generated using Cas9-triggered homologous recombination with a self-excising selection cassette (**Dickinson et al., 2015**). The target sequence was 5′- CTTTCA-GAGCATTGTCACTT**TGG**-3′ with PAM site shown in bold; the guide RNA was cloned into the *eft-3p::Cas9+sgRNA* expression vector pDD162 using this same primer sequence (excluding the PAM). The PAM is 16 nucleotides upstream of the start codon of *inx-9*. Homology arms were amplified by PCR (from N2 genomic DNA, 5′ homology arm forward primer *inx-9* sequences are 5′-gaaataatcga-gatgaaactgtcg-3′ and reverse primer 5′-CAT<u>t</u>ctgtccctttgaacgaaagtg-3′ (with mutation in PAM site shown underlined); 3′ homology arm forward primer *inx-9* sequences 5′-ATGTTTTCTGTTCCATTTC TTACC-3′ and reverse primer 5′-tgagttggactgacatcgag-3′). Note that the beginning of *inx-8* and *inx-*

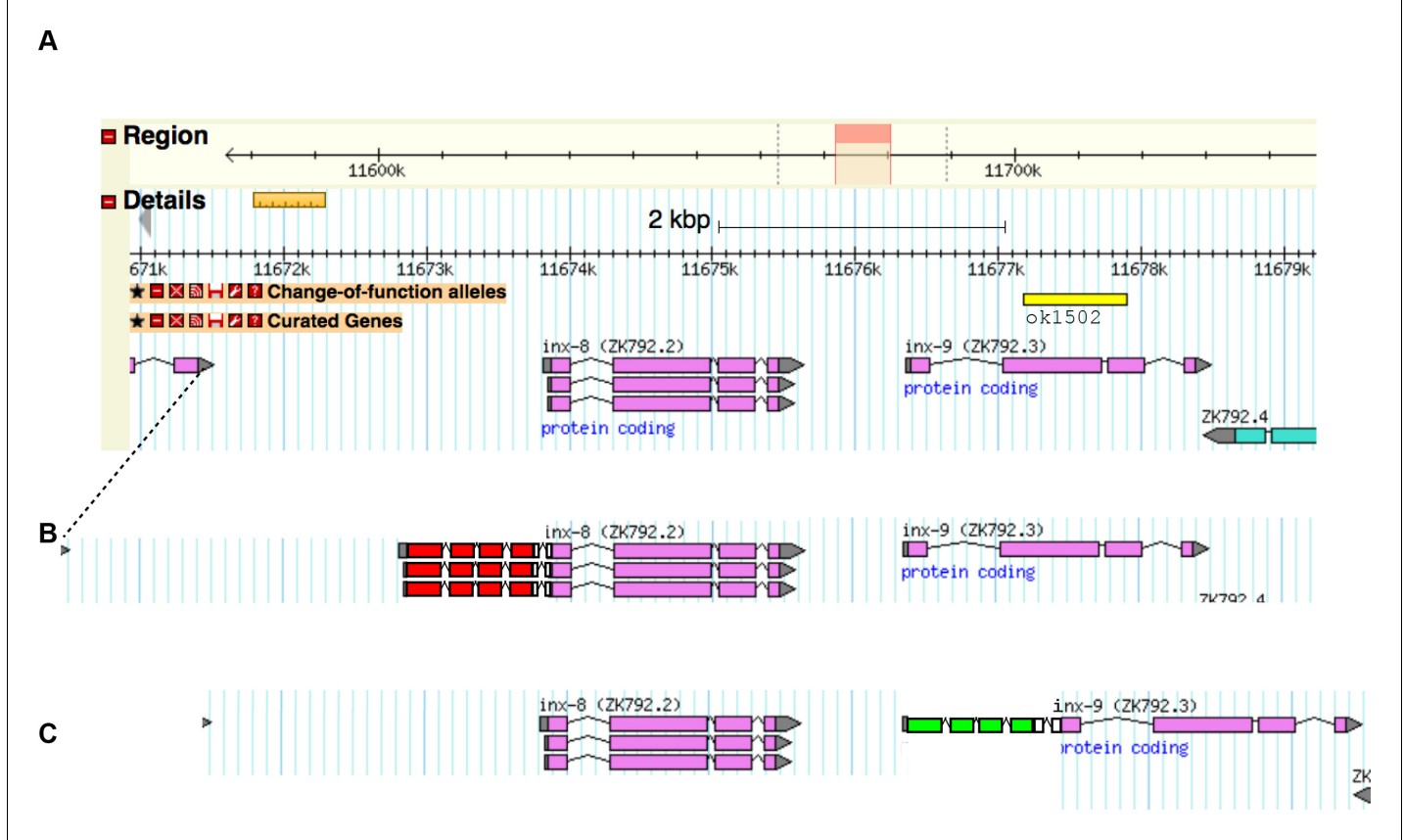

**Scheme 1.** Gene models of the *inx-8/inx-9* locus with alleles annotated. Related to Materials and methods. (**A**) Genomic locus from Wormbase legacy genome browser. Change of function allele *inx-9(ok1502)* (see *Figure 1—figure supplement 1*) annotated with yellow bar. (**B**) Annotation of genome-edited allele *inx-8(qy78) mKate::inx-8* with *mKate* exons shown with red boxes, introns shown with peaked lines, and linker amino acid sequences shown with white boxes. (**C**) Annotation of genome-edited allele *inx-9(qy79) GFP::inx-9* with GFP exons shown with green boxes, introns shown with peaked lines, and linker amino acid sequences shown with white boxes. Endogenous UTRs and regulatory sequences, and their relative positions to their ends of the genes, are preserved.

*9* coding sequences are identical (this is why both sgRNAs target sequences upstream of the coding region). To selectively amplify from *inx-9* for the 3' homology arm, we first used as template a PCR amplicon from N2 genomic DNA including 4.3 kb of the *inx-9* locus using primers that exclude the region of sequence identity at the beginning of the *inx-8* gene (forward primer in *inx-8* downstream exon 5'-GTGACTTCCAAGTTCGTGAGATGGC-3' and reverse primer 5'-gcttgaaaacggtgcggatccagc-3'>800 nt downstream of *inx-9* stop codon). These homology arms were cloned into a plasmid modified from *Dickinson et al., 2015* containing a GFP coding gene, a 54 amino acid linker sequence (including TEV site and ZF1 tag) and a self-excising cassette (SEC) flanked by LoxP sites in an intron of GFP (*Scheme 1C*). The plasmid was coinjected with sgRNA+Cas9 plasmid into N2 animals, and genome-edited animals were selected by hygromycin B treatment and phenotypic identification (roller). The selection cassette excised stochastically and non-rollers were recovered. This tagged protein was not assessed for its function in an *inx-8* mutant background.

The strain containing the *lag-2p::mTagBFP:: PLC$^{\delta PH}$* to mark the DTC, the *mKate2::inx-8(qy78)* to mark Sh1 cells, the germ cell nuclear histone tag *mex-5::H2B::mCherry*, and the germ cell membrane marker *mex-5p::GFP:: PLC$^{\delta PH}$* was maintained as heterozygotes for the germ cell nuclear and germ cell membrane markers, as both are carried in the same Mosci site on chromosome II. While the germ cell membrane maker was helpful in determining whether a germ cell bare region exists, it was ultimately dispensable for inferring germ cell-somatic cell associations in the presence of a germ cell nuclear marker due to the tight packing of germ cells and the close intercalation of the somatic cells.

## Time-lapse imaging

Worms were anesthetized with 0.02% tetramisole in M9 buffer for 15 min prior to mounting on 4% noble agar pads with coverslips sealed with valap and flooded underneath with 0.02% tetramisole to prevent drying (adapted from *Gerhold et al., 2015*, which used 0.04% tetramisole and grooved agarose pads). We found that tricaine and sodium azide, two commonly used worm anesthetics, caused germ cell divisions to cease. Acquisition intervals were set for every five minutes, but the relatively light anesthetic meant that some worm movement still occurred, and pauses to reposition the field of view were occasionally necessary. We found that lowering the laser power of the 402 nm and 488 nm lasers helped decrease spontaneous worm movement and photobleaching. Animals with cell divisions that did not progress were not analyzed.

## Microscopy and image acquisition, processing, and analysis

Confocal DIC and fluorescent images were acquired on an AxioImager A1 microscope (Carl Zeiss) equipped with an EMCCD camera (Hamamatsu Photonics), a 100x or 40x Plan-Apochromat (1.4 NA) objective, and a spinning disc confocal scan head (CSU-10; Yokogawa Electric Corporation) driven by µManager software (*Edelstein et al., 2010*) at 20℃, except *Figure 7D and E*, which were acquired on a Leica DMI8 with an xLIGHT V3 confocal spinning disk head (89 North) with a 63x Plan-Apochromat (1.4 NA) objective and an ORCA-Fusion Gen-III sCMOS camera (Hamamatsu Photonics). Worms were mounted on 4% noble agar pads containing 0.01 M sodium azide for imaging during endpoint experiments. Images were processed with FIJI 2.0 and Photoshop CC (Adobe Systems Inc). Images are displayed as single confocal z-slices or maximum intensity projections generated in FIJI, as noted. Supplemental videos were generated with FIJI. Graphs generated by R and MS Excel were refined using Illustrator CC (Adobe Systems Inc).

## Timed RNAi

We performed timed RNAi by feeding on worms carrying a DTC membrane marker (*lag-2p:: mTagBFP::PLC$^{\delta PH}$*) the *inx-9(qy79)* GFP-tagged allele and the *nasi2(mex-5p::H2BmCherry)* transgene marking germ cell nuclear histones. We did not expose animals to L1 larval RNAi for fear that early knockdown of actin associated proteins might lead to arrest of development (*Ono et al., 2008*; *Ono et al., 2003*). To circumvent the early defects caused by their loss of function, we applied postembryonic RNAi by feeding starting at the early L4 larval stage. By the time RNAi knockdown is achieved, gonad migration is complete. Worms were terminally anesthetized in 0.01 M sodium azide and imaged after 48 hr on RNAi (in the second day of adulthood). Note that these animals are ~24 hr older than most of the other worms shown in the study.

## Measurements of RNAi-treated gonads

Gonad images were measured in FIJI for the length of the longest continuous DTC process, length of the mitotic zone, and distance of distal Sh1 from the distal end of the gonad. Mitotic figures and their positions relative to the fluorescently labeled somatic cells were noted. Germ cells were assessed for their general condition (germ cell number, large germ cells, gaps among germ cells, etc.) and the Sh1 cell was assessed for its condition (normal or stringy). We noted variation in the extent of germ line damage due to *unc-60* RNAi knockdown, with one replicate showing highly abnormal distal germ lines (as noted in *Ono et al., 2008*) with gaps among the misshapen germ cell nuclei presumably caused by the collapse of the rachis and the failure of cytokinesis, which both require proper actin remodeling (*Dorn et al., 2016*; *Priti et al., 2018*). In this replicate, germ cell nuclear size was a more reliable marker of the end of the proliferative zone than nuclear crescent shape. The results shown in *Figure 7* are pooled from three replicates of exposures starting on three consecutive days for a total of 31 control animals and 35 *unc-60* RNAi-treated animals.

## Quantification

Dividing germ cells were identified by chromatin condensations at metaphase and anaphase; their distance from the distal tip of the gonad was measured in FIJI. Distal gonads encompassed by the field of view at 1000x magnification do not always contain the entire progenitor zone, so our estimates of the numbers of undifferentiated germ cells in contact with Sh1 alone is conservative. This distal field of view did contain the DTC-Sh1 interface in every non-RNAi-treated animal in the study.

Germ cell-somatic cell contacts were scored by eye, and typically only germ cells in the superficial layer were counted, as deeper contacts are more ambiguous. Angles of dividing cells, the gonad axis, and the shortest path through the dividing cell from DTC to Sh1 were measured in FIJI and analyzed as shown in *Figure 6* schematic. The angles were expressed as the acute angle between the lines shown. RNAi scoring could not be meaningfully blinded because of the strength of the phenotypes, although the dataset shown in *Figure 7—figure supplement 1* was nominally blinded before acquisition and unblinded after scoring. Sample sizes were evaluated a posteriori for statistical significance.

## Scoring of differentiation markers

Genomically encoded fluorescent markers were used to analyze cell fate asymmetry across the DTC-Sh1 boundary. The marker of a stem-like fate is *sygl-1p::H2B::gfp* (*Kershner et al., 2014*), and we set a threshold in the images to exclude all residual GFP below the expression level of the distal most cells, based on findings that the *sygl-1* gene is actively transcribed in a distal subpopulation but its transcripts perdure in more cells further proximal (*Lee et al., 2016*). We analyzed whether every part of the DTC was in contact with *sygl-1p::H2B::GFP* (+) cells—in most cases, DTCs had long processes that extended proximal to the proximal boundary of *sygl-1p::H2B::GFP* expression. Then we analyzed the region of overlap between Sh1 and *sygl-1p::H2B::GFP* (+) cells, and found it confined to the DTC-Sh1 interface. The marker of the differentiating fate is a *gld-1p::gld-1::GFP* transgene (*ozIs5*; *Brenner and Schedl, 2016*; *Schumacher et al., 2005*; *Seidel et al., 2018*). Looking only at the green channel with the DTC marker and GLD-1::GFP, we examined the farthest distal perinuclear GFP speckles that we could detect in the germ line, and marked their position in the image. We also marked the length of the longest DTC process. We then turned on the red channel and noted the position of the distal extent of Sh1. These data are plotted in *Figure 5D*. The measurements for Sh1 and GLD-1::GFP were made the same way for the RNAi experiment shown in *Figure 7F and G*.

## Statistical analysis

Linear model followed by the Tukey-Kramer HSD test (nmle package in R), prop.test (stats v3.6.0 package), Welch two-sample t-test (t.test, stats v3.6.2 package), Pearson's correlation test (cor.test, stats v3.6.2), or one tailed Wilcoxon Signed Rank tests (stats v3.6.0 package) were performed in R as noted in figure legends and text.

# Acknowledgements

We would like to thank members of the Sherwood Lab, especially Lara Linden-High and Sara Payne (Duke), Ari Pani (UVA), Andres Collazo (Cal Tech), David Greenstein (University of Minnesota), Sarah Crittenden (UW-Madison), Judith Kimble (UW-Madison), and our three reviewers for helpful feedback. We thank Abigail Gerhold (McGill University) and Jane Hubbard (NYU) for sharing strains. Some strains were provided by the CGC, which is funded by NIH Office of Research Infrastructure Programs (P40 OD010440). KLG was supported by postdoctoral fellowship GM121015-01 from the NIGMS. This work was supported by NIGMS R01 GM079320 and NIGMS R35 MIRA GM118049 to DRS.

# Additional information

## Funding

| Funder | Grant reference number | Author |
| --- | --- | --- |
| National Institute of General Medical Sciences | R01 GM079320 | David R Sherwood |
| National Institute of General Medical Sciences | R35 MIRA GM118049 | David R Sherwood |
| National Institute of General Medical Sciences | GM121015-01 | Kacy L Gordon |

The funders had no role in study design, data collection and interpretation, or the decision to submit the work for publication.

## Author contributions
Kacy L Gordon, Conceptualization, Data curation, Formal analysis, Supervision, Validation, Investigation, Visualization, Methodology, Writing - original draft, Project administration, Writing - review and editing; Jay W Zussman, Camille Miller, Investigation, Writing - review and editing; Xin Li, Validation, Investigation, Writing - review and editing; David R Sherwood, Conceptualization, Resources, Supervision, Funding acquisition, Writing - review and editing

## Author ORCIDs
Kacy L Gordon (iD) https://orcid.org/0000-0003-0967-4020
David R Sherwood (iD) http://orcid.org/0000-0003-2245-2334

## Decision letter and Author response
Decision letter https://doi.org/10.7554/eLife.56383.sa1
Author response https://doi.org/10.7554/eLife.56383.sa2

## Additional files
### Supplementary files
• Supplementary file 1. Time-lapse movies analyzed for interface division asymmetry. Related to *Figure 4*.

• Transparent reporting form

### Data availability
Source files for all figure graphs have been provided.

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

# Appendix 1

## Key Resources Table

**Appendix 1—key resources table**

| Reagent type (species) or resource | Designation | Source or reference | Identifiers | Additional information |
|---|---|---|---|---|
| Gene (*C. elegans*) | *inx-8* | wormbase.org | Sequence CELE_ZK792.2 | Encodes gap junction hemichannel subunit |
| Gene (*C. elegans*) | *inx-9* | wormbase.org | Sequence CELE_ZK792.3 | Encodes gap junction hemichannel subunit |
| Gene (*C. elegans*) | *tbb-2* | wormbase.org | Sequence CELE_C36E8.5 | Encodes worm beta tubulin ortholog |
| Gene (*C. elegans*) | *unc-60* | wormbase.org | Sequence CELE_C38C3.5 | Encodes worm cofilin ortholog |
| Gene (*C. elegans*) | *arx-2* | wormbase.org | Sequence CELE_K07C5.1 | Encodes Arp2/3 subunit |
| Strain, strain background (*Escherichia coli*) | HT115(DE3) | Caenorhabditis Genetics Center (CGC) | HT115(DE3) | RNAi feeding strain |
| Genetic reagent (*Escherichia coli*) | Ahringer RNAi Library | Source Bioscience | *C. elegans* RNAi Collection (Ahringer) | |
| Chemical compound, drug | hygromycin B from *Streptomyces hygroscopicus* | Millipore Sigma | CAS 31282-04-9 | |
| Genetic reagent (*C. elegans*) | *unc-119(ed4)* | Paul Sternberg | PS3460 | for transgenic rescue |
| Genetic reagent (*C. elegans*) | *xnSi1(mex-5p::GFP::PLCδ$^{PH}$::nos-2 3'UTR) II; inx-8(qy78(mKate::inx-8)) IV; qyIS546(lag-2p::mTagBFP::PLCδ$^{PH}$)* | *xnSi1* parent from ***Chihara and Nance, 2012*** doi:10.1242/dev.079863 | NK2569 | Can be obtained from K. Gordon lab |
| Genetic reagent (*C. elegans*) | *naSi2(mex-5p::H2B::mCherry::nos-2 3'UTR) II/xnSi1(mex-5p::GFP::PLCδ$^{PH}$::nos-2 3'UTR) II; inx-8(qy78(mKate::inx-8)) IV; qyIS546(lag-2p::mTagBFP::PLCδ$^{PH}$)* | *nasi2* transgene from ***Roy et al., 2018*** doi:10.1534/g3.118.200511; *xnSi1* parent from ***Chihara and Nance, 2012*** doi:10.1242/dev.079863; others this study | NK2570 | Can be obtained from K. Gordon lab |
| Genetic reagent (*C. elegans*) | *inx-8(qy78(mKate::inx-8)) IV; cpIs122(lag-2p::mNeonGreen:: PLCδ$^{PH}$)* | This study | NK2571 | Can be obtained from K. Gordon lab |
| Genetic reagent (*C. elegans*) | *inx-9(qy79(GFP::inx-9)) IV; cpIs91(lag-2p::2x mKate2::PLCδ$^{PH}$::3xHA::tbb-2 3'UTR LoxN) II* | *cpIs91* transgene from ***Gordon et al., 2019*** doi: 10.1016/j.cub.2019.01.056 | NK2572 | Can be obtained from K. Gordon lab |

*Continued on next page*

*Appendix 1—key resources table continued*

| Reagent type (species) or resource | Designation | Source or reference | Identifiers | Additional information |
|---|---|---|---|---|
| Genetic reagent (*C. elegans*) | *naSi2(mex-5p::H2B:: mCherry::nos-2 3'UTR) II; inx-9(qy79(GFP::inx-9)) IV; qyIS546(lag-2p:: mTagBFP::PLC$^{\delta PH}$)* | *nasi2* transgene from ***Roy et al., 2018*** doi: 10.1534/g3.118.200511; others this study | NK2573 | Can be obtained from K. Gordon lab |
| Genetic reagent (*C. elegans*) | *naSi2(mex-5p::H2B:: mCherry::nos-2 3'UTR) II; inx-8(qy78 (mKate::inx-8)) IV; qyIS546(lag-2p:: mTagBFP::PLC$^{\delta PH}$); ojIs1 [pie-1p::GFP::tbb-2 + unc-119(+)]* | *nasi2* transgene from ***Roy et al., 2018*** doi:10.1534/g3.118.200511; *ojIs1* transgene from ***Gerhold et al., 2015*** doi: 10.1016/j.cub.2015.02.054 | NK2574 | Can be obtained from K. Gordon lab |
| Genetic reagent (*C. elegans*) | *naSi2(mex-5p::H2B:: mCherry::nos-2 3'UTR) II; inx-8(qy78 (mKate::inx-8)) IV; arx-2(cas607[arx-2::gfp knock-in]) V* | *nasi2* transgene from ***Roy et al., 2018*** doi:10.1534/g3.118.200511; *arx-2(cas607)* from ***Zhang et al., 2017*** doi: 10.1242/bio.026807 | NK2575 | Can be obtained from K. Gordon lab |
| Genetic reagent (*C. elegans*) | *inx-9(ok1502) IV; inx-8(qy102(mKate::inx-8)) IV; cpIs122(lag-2p::mNeonGreen::PLC$^{\delta PH}$)* | *inx-9(ok1502)* allele from ***Consortium TC elegans DM, 2012*** doi: 10.1534/G3.112.003830 | NK2576 | Can be obtained from K. Gordon lab |
| Genetic reagent (*C. elegans*) | *cpIs91(lag-2p::2x mKate2:: PLC$^{\delta PH}$::3xHA::tbb-2 3'UTR LoxN) II); qSi26(sygl-1p::H2B::GFP::sygl-1 3UTR) II; (inx-8(qy78(mKate::inx-8)) IV; teIs1(oma-1::GFP) IV,* | *qSi26* and *teIs1* from ***Kershner et al., 2014*** doi: 10.1073/pnas.1401861111 | KLG004 | Can be obtained from K. Gordon lab |
| Genetic reagent (*C. elegans*) | *(cpIs122(lag-2p:: mNeonGreen:: PLC$^{\delta PH}$)); inx-8(qy78(mKate::inx-8)) IV; ozIs5(gld-1::GFP)* | *ozIs5* from ***Brenner and Schedl, 2016***; ***Schumacher et al., 2005*** doi: 10.1534/genetics.115.185678 | KLG005 | Can be obtained from K. Gordon lab |
| Sequence-based reagent | *inx-8* 5' arm Forward primer | This study | 5_Nterminx8F | 5'-TGTACGACTG TAGGCAGGCAGGTAG-3' |
| Sequence-based reagent | *inx-8* 5' arm Reverse primer | This study | 5_Nterminx8R | 5'-tctgcaattcgaacccgagtgaagatg-3' mutation in PAM site under-lined |
| Sequence-based reagent | *inx-8* 3' arm Forward primer | This study | 3_Nterminx8F | 5'-ATGTTTTCTGTTCCATTTC TTACCTC-3' |

*Continued on next page*

Appendix 1—key resources table continued

| Reagent type (species) or resource | Designation | Source or reference | Identifiers | Additional information |
|---|---|---|---|---|
| Sequence-based reagent | inx-8 3' arm Reverse primer | This study | 3_Nterminx8R | 5'-CTGTACATC TCCACGGCAACCTCCG-3' |
| Sequence-based reagent | inx-8 sgRNA primer | This study | sg_Nterminx8 | 5'- GCATCTTCACTCGGG TTCGA-3' |
| Sequence-based reagent | inx-9 5' arm Forward primer | This study | 5_Nterminx9F | 5'-gaaataatcgagatgaaactgtcg-3 |
| Sequence-based reagent | inx-9 5' arm Reverse primer | This study | 5_Nterminx9R | 5'-CA Ttctgtccctttgaacgaaagtg-3' mutation in PAM site under-lined |
| Sequence-based reagent | inx-9 3' arm Forward primer | This study | 3_Nterminx9F | 5'-ATGTTTTCTGTTCCATTTC TTACC-3' |
| Sequence-based reagent | inx-9 3' arm Reverse primer | This study | 3_Nterminx9R | 5'-tgagttggactgacatcgag-3' |
| Sequence-based reagent | inx-9 sg RNA primer | This study | sg_Nterminx9 | 5'- CTTTCAGAGCATTGTCAC TT-3' |
| Sequence-based reagent | inx-8 downstream exon Forward primer | This study | Inx8exonF | 5'-GTGACTTCCAAGTTCG TGAGATGGC-3 |
| Sequence-based reagent | downstream of inx-9 stop codon Reverse primer | This study | Inx9downstreamR | 5'-gcttgaaaacggtgcggatccagc-3' |
| Recombinant DNA reagent | arx-2 feeding RNAi clone | Ahringer RNAi library *Kamath and Ahringer, 2003* doi: 10.1016/S1046-2023 (03)00050–1 | HGMP_Location V-7M13 | |
| Recombinant DNA reagent | unc-60 feeding RNAi clone | Vidal RNAi library *Rual et al., 2004* doi:10.1101/gr. 2505604 | RNAi well, GHR-11003@H04 | RNAi well, GHR-11003@H04 |
| Recombinant DNA reagent | modified plasmid for SEC CRISPR strategy | *Dickinson et al., 2015* doi: 10.1534/genetics. 115.178335 | RRID:Addgene_ 132523 | pDD268 |
| Recombinant DNA reagent | eft-3p::Cas9+sgRNA expression vector | *Dickinson et al., 2015* doi: 10.1534/genetics. 115.178335 | RRID:Addgene_ 47549 | pDD162, Addgene plasmid #47549 |
| Recombinant DNA reagent | RNAi empty vector control | Andrew Fire | RRID:Addgene_ 1654 | L4440, Addgene plasmid #1654 |
| Recombinant DNA reagent | unc-119 rescue plasmid | Morris Maduro | pPD#MM016B | |

*Continued on next page*

*Appendix 1—key resources table continued*

| Reagent type (species) or resource | Designation | Source or reference | Identifiers | Additional information |
|---|---|---|---|---|
| Software, algorithm | µManager software v1.4.18 | *Edelstein et al., 2010* doi: 10.1002/0471142727.mb1420s92 | RRID: SCR_016865 | https://micro-manager.org/ |
| Software, algorithm | FIJI 2.0 | *Schindelin et al., 2012* doi:10.1038/nmeth.2019 | RRID: SCR_002285 | https://fiji.sc/ |
| Software, algorithm | Adobe Photoshop CC | Adobe Systems Inc | RRID: SCR_014199 | |
| Software, algorithm | Adobe Illustrator CC | Adobe Systems Inc | RRID: SCR_010279 | |

