## [Decision Letter]

**Acceptance summary:**

This manuscript by Gordon et al., describes germline stem cell divisions in *C. elegans* and provides compelling evidence that GSCs, which have been believed to be maintained at a population level, may divide asymmetrically to create one stem cell and one differentiating cell. This changes a long standing paradigm in the field. The study is well conceived and executed beautifully. The authors addressed the reviewer comments well, and I believe the manuscript is ready for publication.

**Decision letter after peer review:**

[Editors’ note: the authors submitted for reconsideration following the decision after peer review. What follows is the decision letter after the first round of review.]

Thank you for submitting your work entitled "Stem cell niche exit in *C. elegans* via orientation and segregation of daughter cells by a cryptic cell outside the niche" for consideration by *eLife*. Your article has been reviewed by three peer reviewers, including Yukiko M Yamashita as the Reviewing Editor and Reviewer #1, and the evaluation has been overseen by Senior Editor, Utpal Banerjee. The following individual involved in review of your submission has agreed to reveal their identity: Dave Hansen (Reviewer #3).

Our decision has been reached after consultation between the reviewers. Based on these discussions and the individual reviews below, we decided that we cannot consider the publication of your manuscript in its current form.

However, all the reviewers are very enthusiastic about your fascinating observation, and see the potential of breakthrough in the field. Converged concern of reviewers is that the current manuscript does not provide the evidence for 1) Sh1 promoting germ cell differentiation, 2) germ cell division at the border of DTC-Sh1 being truly asymmetric (with respect to the cell fate). Whereas the reviewers felt that the claim can be solidified by toning down the text as well, the overall conclusion was that the manuscript would still require some evidence to support fate asymmetry and fate determination by Sh1 processes.

Whereas these issues can be well addressed within the time frame of normal revision, *eLife*'s policy is to invite revisions only when the revision experiments are expected to be straightforward (unlikely changing major conclusions). Therefore, our decision comes in the form of rejection. However, we would like to convey that the reviewers are very enthusiastic about your work, therefore, if you can address the major comments and submit it as new submission, we would be happy to re-review your paper with the same sets of reviewers.

Specific comments are listed below, but as summarized above, the key point for the revision is to provide the evidence for asymmetric cell fates. As far as the main points are addressed, other points may be addressed simply by modifying the text for accuracy.

Reviewer #1:

In this paper, Gordon and colleagues find that Sh1, a somatic cell population, is located right after the distal tip cell, and encapsulate differentiating germ cells to promote their differentiation. They provide series of very intriguing observations describing how undifferentiated cells physically interact with DTC and Sh1, leading to their proposal that Sh1 creates the microenvironment that promote differentiation. They found that Sh1 cells form thin protrusions that interdigitate with DTC protrusions, where germ cells divide at a high frequency in an oriented manner. This is potentially a significant advancement to the field, but I felt that certain claims are not well supported at this point. The following are the main points that I feel need to be more strongly supported.

Although the authors postulate that Sh1 contact stimulates “niche exit” and thus differentiation of germ cells, they show no evidence that Sh1 indeed promotes differentiation. Although it has been well established that DTC is the niche component, and thus by the extension of logic, Sh1, which exhibits mutually exclusive positional pattern, would correlate with differentiation. However, no evidence is presented that Sh1 indeed plays any roles in differentiation. I don't necessarily think that they have to go too far showing signaling mechanism etc that is elicited by Sh1, but they need to show something that demonstrates Sh1's role in differentiation.

Germ cells that are contacting DTC and Sh1 have disproportionally high incidence of division. They further observed that division orientation of these cells is biased such that one daughter will remain attached to the DTC, and the other attached to Sh1. This is certainly a very interesting observation but it remains unclear why division must be promoted there, and why they divide asymmetrically (promotion of cell proliferation and potentially asymmetric division are two entirely distinct process, which do not have to be tied together). Also, other than the asymmetry in the cell contact (DTC vs. Sh1), no other asymmetry was shown (such as one keeps expressing high Notch vs. the other one low Notch), leaving the functional significance of their observation unclear.

Their only attempt to show the significance of Sh1 encapsulation is in Figure 6 by perturbing actin cytoskeleton. The only outcome they show is reduced proliferation of germ cells at the interface. Again, this echoes with the lack of functional significance in early figures and readers would be left wondering what is the evidence for their conclusion that Sh1 promotes differentiation.

May I also point out to a series of nice work by Tanentzapf lab that showed encapsulation of differentiating cells by somatic cell population in *Drosophila* male germline?

https://www.ncbi.nlm.nih.gov/pubmed/27546574

https://www.ncbi.nlm.nih.gov/pubmed/25503408

In summary, I really like their observations, which potentially provide a breakthrough to the field. However, at this stage, their claims are not strongly supported, I am not asking for whole another paper worth of data (such as which signaling pathway regulates what aspects of their observation etc), but some core data to show the functional significance of Sh1 should be provided.

Reviewer #2:

The manuscript by Gordon et al. addresses the relationship between the spatial structure of the somatic cells interacting with *C. elegans* germline stem cells on the one hand, and the characteristics of germline mitotic divisions on the other hand. Two observations are particularly compelling: 1) the distal tip cell (DTC) and sheath cell pair 1 (Sh1) are much closer together and may interact more actively than realized; 2) the orientation of germ cell division is less random than previously thought, and tends to correlate with the local edges of the somatic cells. The study relies on live imaging that is beautifully executed and still novel in this field, and opens up an important new line of investigation for the field. Weaknesses are that few mechanistic insights are gained, the functional significance is unclear, and the model oversells the results.

Major criticisms:

1) I would expect the manuscript to build on the observation that germ cell divisions are oriented with respect to the local DTC and Sh1 edges, and to provide substantial tests of the regulation and functional significance of this orientation. This is partially done with *unc-60* and *arx-2* RNAi, which extend the gap between the DTC and Sh1 and reduce cell division (as scored in a way that is unclear). A major concern with this experiment is that reduced proliferation could be a result of RNAi directly affecting the germ cells themselves (or a subset of those cells), rather than a result of the enlarged DTC-Sh1 gap – especially since the authors note in the Materials and methods that *unc-60* RNAi leads to severe germ cell morphological defects. It is not clear why soma-specific (or even DTC- or sheath-cell-specific) RNAi was not performed. Furthermore, the molecular changes in germ cells that underlie reduced proliferation and randomized division planes, or potential downstream consequences such as decreased fertility, are not addressed.

2) The story built by the manuscript around the interesting observations is somewhat tautological, does not distinguish cause from effect, and assigns functional purpose that is not substantiated. This is a prominent problem in the following part of the manuscript: "We next tested the hypothesis that by initiating an exclusive association with Sh1 at birth via polarized cell division, a germ cell could directly escape the influence of the DTC niche and embark upon the path to differentiation. […] DTC-Sh1 interface germ cell division, often followed by Sh1 growth over one daughter cell, is the primary mechanism of stem cell niche exit of germ cells."

a) How is "embarking on differentiation" defined? This study does not include markers of germ cell differentiation

b) It appears implied that germ cells need a specific mechanism to exit the stem cell niche. But given the overall movement of germ cells in the distal to proximal direction, why can a germ cell not just passively exit the niche irrespective of what is happening at the DTC-Sh1 interface?

c) The manuscript assumes that germ cell division is regulated by the Sh1-DTC interface. But why could it not be the other way around? (Or could both be downstream of something else?) The Sh1-DTC interface is a very dynamic structure, as reported in this manuscript. Could Sh1 and the DTC not actively remodel their interface in response to germ cell changes that lead up to division?

d) "In most of these cases (56/64) one daughter remained under the DTC or at the interface and the other ended up under or beside Sh1, suggesting a polarized division." If cell division was randomly oriented, would it not often be the case that a division at the Sh1-DTC interface results in one cell closer to the DTC and one cell closer to Sh1? In other words, how strongly do the reported numbers suggest a "polarized division"?

e) Various places where functional purpose is assumed without substantiation: "enwraps the closest daughter to remove it from the niche" (it is not clear why a cell would need to be enwrapped to leave the niche, or that enwrapping plays any role in that exit), "facilitated niche exit", "Sh1 grows over the Sh1-facing daughter, removing it from the niche"

Reviewer #3:

In this manuscript, Gordon et al. describe their analysis of a pair of somatic cells and the effect that these cells have on the stem cell population in the *C. elegans* germ line. This stem cell system is intensively studied and has contributed significantly to our general understanding of how the balance between stem cell self-renewal and differentiation is controlled. This manuscript challenges our understanding of the stem cell population in two significant ways. First, that the first pair of somatic sheath cells are quite far away from the somatic distal tip cell that caps the distal end of the gonad. There has long been thought to be a bare region in between the DTC and sheath cells. Gordon et al., through the use of new markers, demonstrate that the sheath cells are present much more distally than previously appreciated, and that the sheath cells appear to coordinate with the DTC as to the extent of their position. Second, Gordon et al. demonstrate that the plane of germ cell division for the cells at the interface between the DTC and sheath cells is largely perpendicular to the shortest distance between the DTC and sheath cells. Previously, it had been reported and accepted that the orientation of germ cell division is largely random. That it may not be random brings up the possibility of asymmetric cell division, whereas previously it was largely thought to be symmetric. These finding significantly change our understanding of the stem cell population in the *C. elegans* gonad, and stem cell populations in general.

My primary concerns do not have to do with the experiments themselves, but rather with some of the conclusions that were reached. In the Discussion, the authors state "our findings demonstrate that cells outside the niche can play an equal and opposite role to the niche in the balance that maintains stem cell pools with the production of differentiating progeny". At the end of the Introduction, they state "We conclude that Sh1… is an orchestrater of stem cell exit from the niche". While the authors have nicely demonstrated that dividing cells in the DTC-SH1 interface often have one daughter in contact with SH1, and the other with the DTC, they have not directly demonstrated that SH1 helps to regulate the balance between self-renewing and differentiating cells, or that Sh1 helps stem cells exit from the niche. For example, they have not shown that in the absence of SH1 that this balance is affected. Indeed, when they shortened SH1 through RNAi “…the lengths of the progenitor zones were not different between the *unc-60* RNAi-treated animals and control”, suggesting that cells were able to enter meiosis (differentiation) normally. Indeed, McCarter et al., 1997, demonstrated that in the absence of any sheath cells, while proliferation levels are lower and cells do not progress through meiosis normally, cells do enter meiosis (differentiate). Therefore, at a minimum, the sheath cells are not “equal” to the DTC (niche) in balancing maintaining stem cell pools and production of differentiating progeny-ablation of DTC renders cells incapable of self-renewal, while cells are still able to enter meiosis in the absence of sheath cells. The authors can easily address this concern by limiting their conclusions and discussing more explicitly the limitations of their data. A brief description of the McCarter results in the Discussion may be of help.

The authors make a very significant observation in that the orientation of cell division does not appear random in the Sh1-DTC interface. From this they conclude that the cell divisions are asymmetric, which I take to mean that the two daughters are different (perhaps one remaining a stem cell, and the other entering a differentiation pathway). I wonder if this is an accurate conclusion. They show that these daughter cells differ with respect to which somatic cells contact the daughters, and perhaps their position in the gonad arm; however, there isn't really any data that the two daughter cells are different (markers, entered a differentiation pathway, etc.). Perhaps the authors need to be more less strong in the conclusion that this is asymmetric cell division. This could be easily addressed.

[Editors’ note: further revisions were suggested prior to acceptance, as described below.]

Thank you for submitting your article "Stem cell niche exit in *C. elegans* via orientation and segregation of daughter cells by a cryptic cell outside the niche" for consideration by *eLife*. Your article has been reviewed by 3 peer reviewers, and the evaluation has been overseen by a Reviewing Editor and Utpal Banerjee as the Senior Editor. The following individuals involved in review of your submission have agreed to reveal their identity: Yukiko M Yamashita (Reviewer #1); Dave Hansen (Reviewer #3).

The reviewers have discussed the reviews with one another and the Reviewing Editor has drafted this decision to help you prepare a revised submission. Please aim to submit the revised version within two months, but we are happy to extend this timeframe if needed.

This manuscript by Gordon et al., describes germline stem cell divisions in *C. elegans* and provide a compelling evidence that GSCs, which have been believed to be maintained at a population level, may divide asymmetrically to create one stem cell and one differentiating cell. This changes a long standing paradigm in the field. The study is well conceived and executed beautifully.

The reviewers raised several minor points that can all be addressed by textual changes. We anticipate that the manuscript will be accepted without further review. Please provide point-by-point responses together with your revised manuscript.

Reviewer #1:

The manuscript by Gordon et al. reports an important discovery regarding the mode of stem cell division in *C. elegans* germline. It is well established that the distal tip cell (DTC) serves as a niche component of *C. elegans* germline stem cells (GSCs): any germ cells that are associated with the DTC maintains GSC identity, and those that are “pushed away” from DTC-covered region will initiate meiotic program. It has been long believed that GSCs do not divide asymmetrically, and are maintained as a population. This paper fundamentally changes this view by finding that sheath cell (Sh1) generates an alternate pattern as DTC (DTC's cellular processes and Sh1's cellular processes intercalate), and along this boundary between DTC and Sh1, GSCs likely divide asymmetrically.

Their live imaging provides convincing case that GSC divisions, which happens most frequently at the border of DTC and Sh1, are mostly oriented such that one daughter remain associated with DTC, and the other associated with Sh1.

In the previously-reviewed version, the authors did not provide the evidence that this oriented division and daughter cells' association with DTC vs. Sh1 indeed correlates with asymmetric cell fate. In the revised version, they provide convincing evidence that indeed DTC association vs. Sh1 association nicely correlates with stem cell fate (monitored by sygl-1 expression) vs. differentiating cell fate (monitored by gld-1 expression).

Together, this study significantly contributes to our understanding of how the stem cell niche is formed in *C. elegans* gonad. Also, the study reveals that *C. elegans* GSCs divide asymmetrically, completely flipping the previous notion (and the evidence is very strong). I do not have any major concerns at this point, and I predict that this study will be a landmark paper that will be cited for a long time.

Reviewer #2:

I appreciate the efforts the authors made to address the reviewers' comments and strengthen their conclusions. This manuscript reports beautifully-executed experiments, and makes a significant contribution to the *C. elegans* germline field and, more broadly, to the stem cell field. I am not completely sold in on the implications of the term "facilitated niche exit", but the data are solid, the observations are highly novel and impactful, and the authors are well within their right to propose this interpretation.

There is just one major point left that I cannot wrap my head around. The authors state: "we never saw spontaneous growth of Sh1 over a non-dividing cell, nor displacement of germ cells from the niche by distal-to-proximal pushing produced by other cells dividing entirely under the DTC". If that is correct, and if cells can only leave the "DTC region" by a mitotic division with one daughter cell that remains in the DTC region, then division of cells "entirely under the DTC" should cause a continuous increase in the number of cells in that region. Is that really compatible with the data? I think it would be useful for the authors to resolve this point.

Reviewer #3:

As I stated when I first reviewed this paper, Gordon et al. describe their analysis of a pair of somatic cells and the effect that these cells have on the stem cell population in the *C. elegans* germ line. This manuscript challenges our understanding of the stem cell population in two significant ways. First, that the first pair of somatic sheath cells are quite far away from the somatic distal tip cell that caps the distal end of the gonad. There has long been thought to be a bare region in between the DTC and sheath cells. Gordon et al., through the use of new markers, demonstrate that the sheath cells are present much more distally than previously appreciated, and that the sheath cells appear to coordinate with the DTC as to the extent of their position. Second, Gordon et al. demonstrate that the plane of germ cell division for the cells at the interface between the DTC and sheath cells is largely perpendicular to the shortest distance between the DTC and sheath cells. Previously, it had been reported and accepted that the orientation of germ cell division is largely random. That it may not be random brings up the possibility of asymmetric cell division, whereas previously it was largely thought to be symmetric. Since the last submission, the authors have significantly improved the manuscript. Importantly, they have more accurately described the conclusions that can be made based on their data. They have also added more data that supports the model of daughters of asymmetrically divided cells are different. They do this through the use of GLD-1 and SYGL-1. This is a valuable experiment and supports their model; however, it does have its limitations. For example, I am somewhat confused as to their model for GLD-1 expression. As they mention, previous experiments have shown that the sometimes observed steps of increasing GLD-1 accumulation could be due to folds. Other than these dramatic steps, I thought the increase in GLD-1 accumulation is gradual. Therefore, I am not clear as to the boundary of GLD-1 accumulation that they are showing on Figure 5. Is this where GLD-1 is first detected, or is it at one of the steps? If it is where it is first detected, then this seems to contradict previous studies that show GLD-1 accumulation all the way to the distal end of the gonad, though at low levels. If it is at a step, and if these steps are due to folds, then this increase may not be due to the position of SH1. Is there some way that the folds are affected by SH1 position, or vice versa? Therefore, the use of GLD-1 as a marker could use some additional explanation. The interpretation of this data was made more difficult by the absence of a figure legend for Figure 5. It seems like the labeled Figure legend 5 corresponds to new Figure 6.

---

## [Author Response]

[Editors’ note: the authors resubmitted a revised version of the paper for consideration. What follows is the authors’ response to the first round of review.]

Reviewer #1:In this paper, Gordon and colleagues find that Sh1, a somatic cell population, is located right after the distal tip cell, and encapsulate differentiating germ cells to promote their differentiation. They provide series of very intriguing observations describing how undifferentiated cells physically interact with DTC and Sh1, leading to their proposal that Sh1 creates the microenvironment that promote differentiation. They found that Sh1 cells form thin protrusions that interdigitate with DTC protrusions, where germ cells divide at a high frequency in an oriented manner. This is potentially a significant advancement to the field, but I felt that certain claims are not well supported at this point. The following are the main points that I feel need to be more strongly supported.Although the authors postulate that Sh1 contact stimulates “niche exit” and thus differentiation of germ cells, they show no evidence that Sh1 indeed promotes differentiation. Although it has been well established that DTC is the niche component, and thus by the extension of logic, Sh1, which exhibits mutually exclusive positional pattern, would correlate with differentiation. However, no evidence is presented that Sh1 indeed plays any roles in differentiation. I don't necessarily think that they have to go too far showing signaling mechanism etc that is elicited by Sh1, but they need to show something that demonstrates Sh1's role in differentiation.

We appreciate that all reviewers made a similar recommendation, and have since added a new figure (Figure 5) showing the positional relationships among DTC, Sh1, and two germ cell fate markers of stem-like cells (*sygl-1p::H2B::GFP* (Kershner et al., 2014)) and differentiating cells (*gld-1::GFP* (Brenner and Schedl, 2016)), which correlate with the distal and proximal sides of the DTC-Sh1 boundary, respectively. We also show that when we perturb Sh1 position by *unc-60* RNAi, the distal boundary of GLD-1::GFP expression is shifted proximally (Figure 7), suggesting that Sh1 promotes entry into the differentiation program. As Sh1 is not absolutely required for differentiation (see discussion of McCarter et al., 1997 and Killian and Hubbard, 2005), what our data suggest is that Sh1may act to remove germ cells more effectively from the influence of the DTC/Notch signaling through promoting asymmetric cell division and subsequently its enwrapping Sh1-facing daughters might contribute to preventing DTC contact and Notch signaling. We have also now modified the Discussion as follows:

“Instead, we hypothesize that Sh1 promotes differentiation indirectly through its positional exclusion of the DTC from germ cells born from polarized interface divisions (Figure 8). Since these cells lack DTC-contact-mediated Notch signaling from birth, they can derepress mitosis-promoting factors and eventually differentiate (Brenner and Schedl, 2016).”

Germ cells that are contacting DTC and Sh1 have disproportionally high incidence of division. They further observed that division orientation of these cells is biased such that one daughter will remain attached to the DTC, and the other attached to Sh1. This is certainly a very interesting observation but it remains unclear why division must be promoted there, and why they divide asymmetrically (promotion of cell proliferation and potentially asymmetric division are two entirely distinct process, which do not have to be tied together).

We agree that increased proliferation and asymmetry do not need to be tied together, but nevertheless we observe these two phenomena concurrently at the DTC-Sh1 interface. In our field generally there is a conflation of *mitotic fate* and *mitotic division*. While the DTC promotes the mitotic fate through active Notch signaling, it is not yet known what factors stimulate proliferation in germ cells. As it is clear from our data that both the DTC and Sh1 cells are associated with proliferating germ cells, perhaps proliferative signals from both of these somatic cells have an additive effect on the germ cells at the interface. The interface is the only part of the gonad where the soma can create asymmetric cell contacts (Figure 6), and we have now shown that this asymmetry correlates with cell fate asymmetry (Figure 5).

Also, other than the asymmetry in the cell contact (DTC vs. Sh1), no other asymmetry was shown (such as one keeps expressing high Notch vs. the other one low Notch), leaving the functional significance of their observation unclear.

The new Figure 5 with germ cell fate markers helps address this question of asymmetry. We show that the marker of active Notch signaling *sygl-1p::H2B::GFP* (Kershner et al., 2014) is found under the DTC and the marker of early differentiation *gld-1::GFP* (Brenner and Schedl, 2016) is found under Sh1.

Their only attempt to show the significance of Sh1 encapsulation is in Figure 6 by perturbing actin cytoskeleton. The only outcome they show is reduced proliferation of germ cells at the interface. Again, this echoes with the lack of functional significance in early figures and readers would be left wondering what is the evidence for their conclusion that Sh1 promotes differentiation.

In addition to discovering that markers of germ cell fate *sygl-1p::H2B::GFP* (Kershner et al., 2014) and *gld-1::GFP* (Brenner and Schedl, 2016) segregate at the DTC-Sh1 boundary, we also tested the effect of *unc-60* RNAi on the boundary of GLD-1::GFP. This experiment is technically challenging, as early RNAi treatments with *unc-60* cause more severe systemic defects in the worm, but late exposure to the RNAi was not sufficient to cause any defect by the 24 hours after L4 time point that we characterized the GLD-1::GFP boundary for. Nevertheless, we present data in Figure 7 that shows that the Sh1 phenotype caused by 40+ hours on *unc-60* RNAi—a stringy and proximally displaced distal edge—is correlated with a proximal displacement of GLD-1::GFP, the differentiation marker, which provides evidence that the Sh1 cell promotes germ cell differentiation.

May I also point out to a series of nice work by Tanentzapf lab that showed encapsulation of differentiating cells by somatic cell population in *Drosophila* male germline?https://www.ncbi.nlm.nih.gov/pubmed/27546574https://www.ncbi.nlm.nih.gov/pubmed/25503408

Thank you for pointing out this important literature. It is certainly relevant to the question of how niche-adjacent cells can limit the influence of the stem cell niche. We have now cited this work:

“Some niche-adjacent cells limit the range of niche cues to restrict the stem cell population, like the *Drosophila* testis cyst cells (Fairchild et al., 2016, 2015).”

In summary, I really like their observations, which potentially provide a breakthrough to the field. However, at this stage, their claims are not strongly supported, I am not asking for whole another paper worth of data (such as which signaling pathway regulates what aspects of their observation etc), but some core data to show the functional significance of Sh1 should be provided.

Thank you for recognizing the potential of our findings to open up new areas of inquiry in a canonical stem cell niche model system. We took to heart the suggestion to pursue markers of germ cell fate as a way to directly interrogate the relationship between germ cell differentiation and somatic cell contact (see Figures 5 and 7, and their associated Results sections).

Reviewer #2:The manuscript by Gordon et al. addresses the relationship between the spatial structure of the somatic cells interacting with *C. elegans* germline stem cells on the one hand, and the characteristics of germline mitotic divisions on the other hand. Two observations are particularly compelling: 1) the distal tip cell (DTC) and sheath cell pair 1 (Sh1) are much closer together and may interact more actively than realized; 2) the orientation of germ cell division is less random than previously thought, and tends to correlate with the local edges of the somatic cells. The study relies on live imaging that is beautifully executed and still novel in this field, and opens up an important new line of investigation for the field. Weaknesses are that few mechanistic insights are gained, the functional significance is unclear, and the model oversells the results.

We thank the reviewer for identifying the most important findings of our study, and for the kind words about the live imaging experiments. We especially are grateful for the moderation the reviewer urges in not over-interpreting the findings we have made. We have now toned-down our interpretation (see following responses) and provided new experiments that more strongly support the relationship among the DTC, Sh1, and the proliferative germ cells.

Major criticisms:1) I would expect the manuscript to build on the observation that germ cell divisions are oriented with respect to the local DTC and Sh1 edges, and to provide substantial tests of the regulation and functional significance of this orientation. This is partially done with unc-60 and arx-2 RNAi, which extend the gap between the DTC and Sh1 and reduce cell division (as scored in a way that is unclear). A major concern with this experiment is that reduced proliferation could be a result of RNAi directly affecting the germ cells themselves (or a subset of those cells), rather than a result of the enlarged DTC-Sh1 gap – especially since the authors note in the Materials and methods that unc-60 RNAi leads to severe germ cell morphological defects. It is not clear why soma-specific (or even DTC- or sheath-cell-specific) RNAi was not performed. Furthermore, the molecular changes in germ cells that underlie reduced proliferation and randomized division planes, or potential downstream consequences such as decreased fertility, are not addressed.

We thank the reviewer for pointing out the need for improved quantification and description of the germ cell division phenotype that we describe upon *unc-60* RNAi treatment in Figure 7. We have presented the data in a histogram like that shown in Figure 3. This analysis revealed that the total number and positions of the DTC-associated germ cells is relatively similar (among RNAi-treated animals with germ cell divisions). The DTC-Sh1 boundary divisions that are almost entirely absent after RNAi treatment account for the overall diminished proliferation observed after RNAi treatment. These results strongly supporting the notion that *unc-60* RNAi is not generally eliminating cell division but rather that the reduction we are seeing is a result of the loss of Sh1 processes at the DTC-Sh1 boundary (see Results section):

“We assessed the positions and somatic cell contacts of these dividing cells as in Figure 3E. The numbers and positions of actively dividing germ cells in contact with the DTC alone are similar in the control and unc-60 RNAi-treated animals (Figure 7C yellow solid and hashed), indicating that the RNAi treatment did not dramatically affect divisions of germ cells in the DTC plexus. However, dividing germ cells at the interface were almost entirely absent because the interface was largely eliminated (compare solid pink to hashed pink, Figure 7C). If we focus on the animals with active divisions (ignoring the 40% of RNAi-treated animals that lack dividing cells entirely), the control (n = 46/26 or 1.77 DTC divisions/animal) and treatment (n = 38/21 or 1.81 DTC divisions/animal) groups have roughly equivalent average numbers of DTC-associated divisions. However, the number of DTC-Sh1 interface divisions is dramatically different between control (n = 23/26 or 0.88 DTC-Sh1 divisions/animal) and treatment (n = 5/21 or 0.24 DTC-Sh1 divisions/animal).”

We think the presence of mitotic figures and a normal sized mitotic zone suggests that germ cell autonomous *unc-60* RNAi defects are not the driving force behind the defects we measured. Downstream fertility defects were not analyzed because of embryonic effects of maternal *unc-60* RNAi.

2) The story built by the manuscript around the interesting observations is somewhat tautological, does not distinguish cause from effect, and assigns functional purpose that is not substantiated. This is a prominent problem in the following part of the manuscript: "We next tested the hypothesis that by initiating an exclusive association with Sh1 at birth via polarized cell division, a germ cell could directly escape the influence of the DTC niche and embark upon the path to differentiation. […] DTC-Sh1 interface germ cell division, often followed by Sh1 growth over one daughter cell, is the primary mechanism of stem cell niche exit of germ cells."a) How is "embarking on differentiation" defined? This study does not include markers of germ cell differentiation

“Embarking on the path to differentiation” is a commonly used phrase in the *C. elegans* germ line literature (see Crittenden, 2006, Kimble and Crittenden, 2005 Wormbook chapter), but we thank the reviewer for pointing out that it risks being confusing to a broader audience and we have removed it. It refers to cells of the progenitor zone that are no longer stem-like (have lost contact with the DTC), but may still divide mitotically. These cells will go on to differentiate, but they are not differentiated yet:

“We next asked how germ cells could escape this contact-mediated signal from the DTC niche and become able to differentiate. We hypothesized that these cells might either lose DTC contact and gain Sh1 associations by moving, initiate an exclusive association with Sh1 at birth via polarized cell division, or both.”

Low levels of GLD-1::GFP (Brenner and Schedl, 2016) mark this cell fate, which we now show in Figure 5. The suggestion to look at markers of cell fate was very helpful.

b) It appears implied that germ cells need a specific mechanism to exit the stem cell niche. But given the overall movement of germ cells in the distal to proximal direction, why can a germ cell not just passively exit the niche irrespective of what is happening at the DTC-Sh1 interface?

The reviewer is correct that it is theoretically possible that germ cells are able to leave the niche passively. However, over a combined ~12 hours of time-lapse imaging of 468 cells at the DTC-Sh1 interface, we never saw a cell leave contact with the DTC without dividing at the interface and one daughter becoming associated with Sh1. Furthermore, we never saw a cell divide in contact with the DTC alone and give rise to a daughter cell that ended up out of DTC contact. This is described in the Results section:

“These dividing interface cells were the only cells (out of 468 total interface cells examined) that changed their germ cell-somatic cell contacts during observation. Similarly, out of the 55 scored cell divisions that occurred in contact with the DTC alone (Supplementary file 1), none of the daughters appeared to leave the niche, so division alone is not sufficient to displace a daughter cell out of the niche.”

We hypothesize that niche exit is one of many biological events that is overdetermined, or for which there is an added layer of regulatory control.

c) The manuscript assumes that germ cell division is regulated by the Sh1-DTC interface. But why could it not be the other way around? (Or could both be downstream of something else?) The Sh1-DTC interface is a very dynamic structure, as reported in this manuscript. Could Sh1 and the DTC not actively remodel their interface in response to germ cell changes that lead up to division?

We thank the reviewer for this very thoughtful comment. Indeed, we favor the hypothesis that Sh1 growth is stimulated by the precursors of cytokinesis. The fact that a germ cell’s spindles orient to the DTC and Sh1 points of contact before it divides suggests that the interaction between the germ cells and Sh1 initiates before division. We present these data and their interpretation in the following Results section:

“Spindle orientation was observed across the DTC-Sh1 interface prior to division, which is consistent with a model in which the DTC-Sh1 interface dictates the asymmetric division (as opposed to the completed cell division later triggering remodeling at the DTC-Sh1 interface).”

d) "In most of these cases (56/64) one daughter remained under the DTC or at the interface and the other ended up under or beside Sh1, suggesting a polarized division." If cell division was randomly oriented, would it not often be the case that a division at the Sh1-DTC interface results in one cell closer to the DTC and one cell closer to Sh1? In other words, how strongly do the reported numbers suggest a "polarized division"?

We thank the reviewer for this probing question, and agree that the positional data alone is suggestive but not definitive evidence for oriented division. Stronger evidence comes from investigating the spindle positions of dividing cells at the interface with marked DTC, Sh1, germ cell histones, and tubulin (Figure 6). If the spindles were *not* polarized between the DTC and Sh1, then we would expect our rosette plot in Figure 6E to resemble the plot in Figure 6C, which shows no relationship between the spindle orientation and the gonad axis. Instead, the division angles to the DTC-Sh1 axis are significantly smaller (more aligned).

e) Various places where functional purpose is assumed without substantiation: "enwraps the closest daughter to remove it from the niche" (it is not clear why a cell would need to be enwrapped to leave the niche, or that enwrapping plays any role in that exit), "facilitated niche exit", "Sh1 grows over the Sh1-facing daughter, removing it from the niche"

Thank you for this helpful observation. Our fate markers give us additional evidence niche exit occurs when a germ cell is in contact with Sh1 and not the DTC, we have modified the language to remove any implication of goal-directed behavior by cells, or to reach beyond the functions we have evidence to support.

The first quoted phrase has been replaced:

“…we made the surprising discovery that some cells in the *C. elegans* germ line stem cell niche divide in an asymmetric manner towards a niche-adjacent cell. Germ cells that contact this cell instead of the niche have begun to differentiate”

The last quoted phrase has been replaced:

“Sh1 actively grows over and between daughter cells that divide at the interface in an Arp2/3- and cofilin-dependent manner, which strengthens the association between one daughter cell and the sheath.”

Additionally, we changed the Abstract that read: “and Sh1 grew over the Sh1-facing daughter, removing it from the niche,” to read as follows: “…and Sh1 grew over the Sh1-facing daughter.”

The word “remove” no longer appears in the manuscript.

“facilitated niche exit” appeared two times in the Discussion section. We deleted one, and the other has been changed as follows:

“This discovery raises the possibility that facilitated niche exit is part of the repertoire of cellular behaviors that regulate the balance between stem cell renewal and differentiation and a new player to a canonical stem cell niche system.”

We were surprised to find that this concept—a cell or structure that removes differentiating progeny from a stem cell niche—is not represented in the literature. While this first paper may not prove that the Sh1 cell is performing this role, we do feel it is a justified point of discussion to name the phenomenon.

Reviewer #3:In this manuscript, Gordon et al. describe their analysis of a pair of somatic cells and the effect that these cells have on the stem cell population in the *C. elegans* germ line. This stem cell system is intensively studied and has contributed significantly to our general understanding of how the balance between stem cell self-renewal and differentiation is controlled. This manuscript challenges our understanding of the stem cell population in two significant ways. First, that the first pair of somatic sheath cells are quite far away from the somatic distal tip cell that caps the distal end of the gonad. There has long been thought to be a bare region in between the DTC and sheath cells. Gordon et al., through the use of new markers, demonstrate that the sheath cells are present much more distally than previously appreciated, and that the sheath cells appear to coordinate with the DTC as to the extent of their position. Second, Gordon et al. demonstrate that the plane of germ cell division for the cells at the interface between the DTC and sheath cells is largely perpendicular to the shortest distance between the DTC and sheath cells. Previously, it had been reported and accepted that the orientation of germ cell division is largely random. That it may not be random brings up the possibility of asymmetric cell division, whereas previously it was largely thought to be symmetric. These finding significantly change our understanding of the stem cell population in the *C. elegans* gonad, and stem cell populations in general.My primary concerns do not have to do with the experiments themselves, but rather with some of the conclusions that were reached. In the Discussion, the authors state "our findings demonstrate that cells outside the niche can play an equal and opposite role to the niche in the balance that maintains stem cell pools with the production of differentiating progeny".

We appreciate greatly that the reviewer recognizes the importance of our work to the field of stem cell biology. In this statement, they are correct that “equal” is an overstatement and this sentence has been removed.

At the end of the Introduction, they state "We conclude that Sh1… is an orchestrater of stem cell exit from the niche".

We thank the reviewer for highlighting this language and in the interest of not overinterpreting our results (also mentioned by the other reviewers), we have removed the reference to “orchestration” at the end of the Introduction.

While the authors have nicely demonstrated that dividing cells in the DTC-SH1 interface often have one daughter in contact with SH1, and the other with the DTC, they have not directly demonstrated that SH1 helps to regulate the balance between self-renewing and differentiating cells, or that Sh1 helps stem cells exit from the niche. For example, they have not shown that in the absence of SH1 that this balance is affected. Indeed, when they shortened SH1 through RNAi “…the lengths of the progenitor zones were not different between the unc-60 RNAi-treated animals and control”, suggesting that cells were able to enter meiosis (differentiation) normally. Indeed, McCarter et al., 1997, demonstrated that in the absence of any sheath cells, while proliferation levels are lower and cells do not progress through meiosis normally, cells do enter meiosis (differentiate). Therefore, at a minimum, the sheath cells are not “equal” to the DTC (niche) in balancing maintaining stem cell pools and production of differentiating progeny-ablation of DTC renders cells incapable of self-renewal, while cells are still able to enter meiosis in the absence of sheath cells. The authors can easily address this concern by limiting their conclusions and discussing more explicitly the limitations of their data. A brief description of the McCarter results in the Discussion may be of help.

We were inspired by this reviewer’s thoughtful comment to directly test GLD-1::GFP expression (a meiosis-promoting factor that is repressed in stem cells by active Notch signaling) in the *unc-60* RNAi-treated worms. We found a significant but minor shift away from the distal end (~8 μm) in treated vs. control animals. The magnitude of the change may explain why the overall length of the progenitor zone is not different between controls and RNAi treated animals.

We agree that McCarter et al., 1997 is the most comprehensive previous work done on Sh1 and that it demonstrates germ cells are able to differentiate in the absence of Sh1, meaning Sh1 is not required for differentiation as the DTC is for the mitotic fate. We have removed the phrasing “equal” and have expanded the Discussion of McCarter et al., 1997:

“Our results show that Sh1 is not required for germ cell differentiation—GLD-1::GFP is visible in germ cells in the gap between the DTC and Sh1 when Sh1 is shifted proximally—much as we would expect given that differentiation is still observed in the reduced germ cell pools of sheath-ablated animals (McCarter et al., 1997 and Killian and Hubbard, 2005). Instead, we hypothesize that Sh1 promotes differentiation indirectly through its positional exclusion of the DTC from germ cells born from polarized interface divisions (Figure 8).”

The authors make a very significant observation in that the orientation of cell division does not appear random in the Sh1-DTC interface. From this they conclude that the cell divisions are asymmetric, which I take to mean that the two daughters are different (perhaps one remaining a stem cell, and the other entering a differentiation pathway). I wonder if this is an accurate conclusion. They show that these daughter cells differ with respect to which somatic cells contact the daughters, and perhaps their position in the gonad arm; however, there isn't really any data that the two daughter cells are different (markers, entered a differentiation pathway, etc.). Perhaps the authors need to be more less strong in the conclusion that this is asymmetric cell division. This could be easily addressed.

We appreciate this comment, and it agrees with the other reviewers’ comments that more experimental support was needed to make this conclusion. We have now included two germ cell fate markers, one for stem-like cells (*sygl-1p::H2B::*GFP, Kershner et al., 2014) and one for cells that will differentiate (*gld-1::GFP,* Brenner and Schedl, 2016). The data are shown in Figure 5, and are consistent with our cell biological studies of asymmetry across the DTC-Sh1 interface. In animals in the first day of adulthood, all of the GLD-1::GFP positive germ cells are associate with Sh1, suggesting germ cells associating with the Sh1 cell have left the niche and begun to differentiate. Additionally, all of the sygl-1p::H2B::GFP positive cells are limited to the region distal to the DTC-Sh1 interface, inclusive, and do not correlate with long DTC processes.

[Editors’ note: what follows is the authors’ response to the second round of review.]

Reviewer #2:I appreciate the efforts the authors made to address the reviewers' comments and strengthen their conclusions. This manuscript reports beautifully-executed experiments, and makes a significant contribution to the *C. elegans* germline field and, more broadly, to the stem cell field. I am not completely sold in on the implications of the term "facilitated niche exit", but the data are solid, the observations are highly novel and impactful, and the authors are well within their right to propose this interpretation.There is just one major point left that I cannot wrap my head around. The authors state: "we never saw spontaneous growth of Sh1 over a non-dividing cell, nor displacement of germ cells from the niche by distal-to-proximal pushing produced by other cells dividing entirely under the DTC". If that is correct, and if cells can only leave the "DTC region" by a mitotic division with one daughter cell that remains in the DTC region, then division of cells "entirely under the DTC" should cause a continuous increase in the number of cells in that region. Is that really compatible with the data? I think it would be useful for the authors to resolve this point.

Thank you for the astute observation—the number of DTC-alone associated germ cells does indeed increase dramatically from the end of larval gonad elongation through the first days of egg laying.

The reason we do not directly address this question is that there is still somewhat surprising disagreement about the cell cycle length of mitotic germ cells in the adult. The cell cycle length in the adult proliferative zone has been estimated at 5-6 h (Rosu and Cohen-Fix, 2017; Chiang et al., 2015), to 9-12 h (Fox and Schedl, 2015), up to 16-24 hours (Crittenden et al., 2006). My data do not directly bear on this question, so I decided to avoid commenting on the issue in the manuscript.

However, I can take these published estimates and some data I have at home on hard drives to try to answer the question.

I find an average of 10 germ cells under the DTC alone in animals of the last larval stage, L4 (this is unpublished data from a strain imaged at the stage shown in Figure 2 “end migration”).

A day later an average of 34 germ cells are under the DTC alone (This number comes from doubling the 17 germ cells on average shown in Figure 3B “DTC-alone”, since we imaged only the superficial half of the gonad; it’s a rough estimate admittedly).

If we have 10 cells under the DTC alone x doubling rate of 2/12 hours x 24 hours = 40, which is close enough to an average of 34.

We can apply the doubling rate again.

34 cells under the DTC alone x doubling 2/12 hours x 24 hours = 136.

Counting the number of germ cells under the DTC of worms aged 48 hours after the L4 stage for two different strains (one with the DTC and germ cell histones marked, and one with GFP::INX-9 and germ cell histones marked), we get averages of 145 and 125 germ cells, respectively (this is also unpublished data, counted in a dozen full-thickness Z-stacks I made as in Author response image 1; two different projections were made through the top and bottom halves of the gonad to avoid double counting or losing cells by projecting them onto one another). The estimate of 136 falls right into this observed range.

**Author response image 1. sa2fig1:** 

Of course these calculations assume all germ cells divide at the same rate, which we do not believe is true, and it assumes a division rate in the middle of the pack of estimates, which may or may not ultimately be accurate. Actual cell cycles may also vary among the DTC-associated germ cells either over time or spatially within the niche. In fact, as we note in the manuscript, the peak in cell division frequency observed by Maciejowski et al., 2006, overlaps the region of the DTC-Sh1 interface, with a lower rate of division distal to this peak, so cells under the DTC alone may correspond to those that were found in a row-by-row analysis by Maciejowski et al., 2006, to have a lower frequency of division/longer cell cycle length.

Additionally, Chiang et al., 2015, find evidence for a population that balances longer lived, slower cycling distal stem cells with faster dividing progeny that subsequently differentiate, with about a ~1.5-2 fold difference between the slower-cycling cells at the distal end and those more proximal mitotic cells closer to ~10 germ cell diameters from the distal end of the gonad (this corresponds to the neighborhood of the DTC-Sh1 interface).

Finally, remodeling at the distal edge of Sh1 combined with proliferation of DTC-associated germ cells must sometimes lead to a daughter of a DTC-only-associated germ cell landing at the DTC-Sh1 interface, where its subsequent divisions will result in daughters leaving the niche. Longer-term time-lapsing will eventually shed light onto these dynamics.

Taken together, it does appear feasible that germ cells born under the DTC-alone can remain associated with the DTC for the medium and slow estimates of germ cell proliferation rate. Future studies will determine whether these cells ever contribute to gamete formation, if there are certain environmental conditions that “tap into” this reservoir of distal germ cells, or if this is perhaps a hedge against germ cell death of progenitors outside the niche. Alternatively, as we expand our long-term imaging technique, we may see that some germ cells do indeed leave the niche without dividing, albeit at a rate that is too infrequent to have captured with our time-lapsing so far.